# Improving Robustness with Adaptive Weight Decay

**Amin Ghiasi, Ali Shafahi, Reza Ardekani**
Apple
Cupertino, CA, 95014
{mghiasi2, ashafahi, rardekani} @apple.com

## Abstract

We propose adaptive weight decay, which automatically tunes the hyper-parameter for weight decay during each training iteration. For classification problems, we propose changing the value of the weight decay hyper-parameter on the fly based on the strength of updates from the classification loss (i.e., gradient of cross-entropy), and the regularization loss (i.e., $\ell_2$-norm of the weights). We show that this simple modification can result in large improvements in adversarial robustness — an area which suffers from robust overfitting — without requiring extra data across various datasets and architecture choices. For example, our reformulation results in 20% relative robustness improvement for CIFAR-100, and 10% relative robustness improvement on CIFAR-10 comparing to the best tuned hyper-parameters of traditional weight decay resulting in models that have comparable performance to SOTA robustness methods. In addition, this method has other desirable properties, such as less sensitivity to learning rate, and smaller weight norms, which the latter contributes to robustness to overfitting to label noise, and pruning.

## 1 Introduction

Deep Neural Networks (DNNs) have exceeded human capability on many computer vision tasks. Due to their high capacity for memorizing training examples (Zhang et al., 2021), DNN generalization heavily relies on the training algorithm. To reduce memorization and improve generaliazation, several approaches have been taken including regularization and augmentation. Some of these augmentation techniques alter the network input (DeVries & Taylor, 2017; Chen et al., 2020; Cubuk et al., 2019, 2020; Müller & Hutter, 2021), some alter hidden states of the network (Srivastava et al., 2014; Ioffe & Szegedy, 2015; Gastaldi, 2017; Yamada et al., 2019), some alter the expected output (Warde-Farley & Goodfellow, 2016; Kannan et al., 2018), and some affect multiple levels (Zhang et al., 2017; Yun et al., 2019; Hendrycks et al., 2019b). Typically, augmentation methods aim to enhance generalization by increasing the diversity of the dataset. The utilization of regularizers, such as weight decay (Plaut et al., 1986; Krogh & Hertz, 1991), serves to prevent overfitting by eliminating solutions that solely memorize training examples and by constraining the complexity of the DNN. Regularization methods are most beneficial in areas such as adversarial robustness, and noisy-data settings – settings which suffer from catastrophic overfitting. In this paper, we revisit weight decay; a regularizer mainly used to avoid overfitting.

The rest of the paper is organized as follows: In Section 2, we revisit tuning the weight decay hyper-parameter to improve adversarial robustness and introduce Adaptive Weight Decay. Also in Section 2, through extensive experiments on various image classification datasets, we show that adversarial training with Adaptive Weight Decay improves both robustness and natural generalization compared to traditional non-adaptive weight decay. Next, in Section 3, we briefly mention other potential applications of Adaptive Weight Decay to network pruning, robustness to sub-optimal learning-rates, and training on noisy labels.

37th Conference on Neural Information Processing Systems (NeurIPS 2023).

# 2 Adversarial Robustness

DNNs are susceptible to adversarial perturbations (Szegedy et al., 2013; Biggio et al., 2013). In the adversarial setting, the adversary adds a small imperceptible noise to the image, which fools the network into making an incorrect prediction. To ensure that the adversarial noise is imperceptible to the human eye, usually noise with bounded $\ell_p$-norms have been studied (Sharif et al., 2018). In such settings, the objective for the adversary is to maximize the following loss:

$$\max_{|\delta|_p \leq \epsilon} Xent(f(x + \delta, w), y),\tag{1}$$

where $Xent$ is the Cross-entropy loss, $\delta$ is the adversarial perturbation, $x$ is the clean example, $y$ is the ground truth label, $\epsilon$ is the adversarial perturbation budget, and $w$ is the DNN paramater.

A multitude of papers concentrate on the adversarial task and propose methods to generate robust adversarial examples through various approaches, including the modification of the loss function and the provision of optimization techniques to effectively optimize the adversarial generation loss functions (Goodfellow et al., 2014; Madry et al., 2017; Carlini & Wagner, 2017; Izmailov et al., 2018; Croce & Hein, 2020a; Andriushchenko et al., 2020). An additional area of research centers on mitigating the impact of potent adversarial examples. While certain studies on adversarial defense prioritize approaches with theoretical guarantees (Wong & Kolter, 2018; Cohen et al., 2019), in practical applications, variations of adversarial training have emerged as the prevailing defense strategy against adversarial attacks (Madry et al., 2017; Shafahi et al., 2019; Wong et al., 2020; Rebuffi et al., 2021; Gowal et al., 2020). Adversarial training involves on the fly generation of adversarial examples during the training process and subsequently training the model using these examples. The adversarial training loss can be formulated as a min-max optimization problem:

$$\min_{w} \max_{|\delta|_p \leq \epsilon} Xent(f(x + \delta, w), y),\tag{2}$$

## 2.1 Robust overfitting and relationship to weight decay

Adversarial training is a strong baseline for defending against adversarial attacks; however, it often suffers from a phenomenon referred to as *Robust Overfitting* (Rice et al., 2020). Weight decay regularization, as discussed in 2.1.1, is a common technique used for preventing overfitting.

### 2.1.1 Weight Decay

Weight decay encourages weights of networks to have smaller magnitudes (Zhang et al., 2018) and is widely used to improve generalization. Weight decay regularization can have many forms (Loshchilov & Hutter, 2017), and we focus on the popular $\ell_2$-norm variant. More precisely, we focus on classification problems with cross-entropy as the main loss – such as adversarial training – and weight decay as the regularizer, which was popularized by Krizhevsky et al. (2017):

$$Loss_w(x, y) = Xent(f(x, w), y) + \frac{\lambda_{wd}}{2}\|w\|_2^2,\tag{3}$$

where $w$ is the network parameters, $(x, y)$ is the training data, and $\lambda_{wd}$ is the weight-decay hyper-parameter. $\lambda_{wd}$ is a crucial hyper-parameter in weight decay, determining the weight penalty compared to the main loss (e.g., cross-entropy). A small $\lambda_{wd}$ may cause overfitting, while a large value can yield a low weight-norm solution that poorly fits the training data. Thus, selecting an appropriate $\lambda_{wd}$ value is essential for achieving an optimal balance.

### 2.1.2 Robust overfitting phenomenon revisited

To study robust overfitting, we focus on evaluating the $\ell_\infty$ adversarial robustness on the CIFAR-10 dataset while limiting the adversarial budget of the attacker to $\epsilon = 8$ – a common setting for evaluating robustness. For these experiments, we use a WideResNet 28-10 architecture (Zagoruyko & Komodakis, 2016) and widely adopted PGD adversarial training (Madry et al., 2017) to solve the adversarial training loss with weight decay regularization:

$$\min_{w} \big( \max_{|\delta|_\infty \leq 8} Xent(f(x + \delta, w), y) + \frac{\lambda_{wd}}{2}\|w\|_2^2 \big),\tag{4}$$

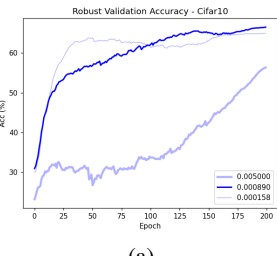 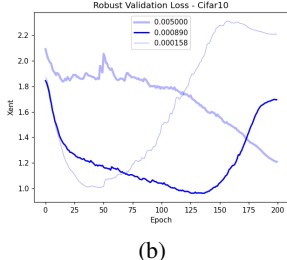 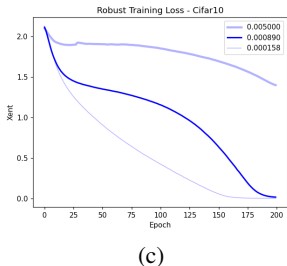

|     |     |     |
|:---:|:---:|:---:|
| (a) | (b) | (c) |

Figure 1: Robust validation accuracy (a) and validation loss (b) and training loss (c) on CIFAR-10 subsets. $\lambda_{wd} = 0.00089$ is the best performing hyper-parameter we found by doing a grid-search. The other two hyper-parameters are two points from our grid-search, one with larger and the other with smaller hyper-parameter for weight decay. The thickness of the plot-lines correspond to the magnitude of the weight-norm penalties. As it can be seen by (a) and (b), networks trained by small values of $\lambda_{wd}$ suffer from robust-overfitting, while networks trained with larger values of $\lambda_{wd}$ do not suffer from robust overfitting but the larger $\lambda_{wd}$ further prevents the network from fitting the data (c) resulting in reduced overall robustness.

We reserve 10% of the training examples as a held-out validation set for early stopping and checkpoint selection. In practice, to solve eq. 4, the network parameters $w$ are updated after generating adversarial examples in real-time using a 7-step PGD adversarial attack. We train for 200 epochs, using an initial learning-rate of 0.1 combined with a cosine learning-rate schedule. Throughout training, at the end of each epoch, the robust accuracy and robustness loss (i.e., cross-entropy loss of adversarial examples) are evaluated on the validation set by subjecting the held-out validation examples to a 3-step PGD attack. For further details, please refer to A.1.

To further understand the robust overfitting phenomenon in the presence of weight decay, we train different models by varying the weight-norm hyperparameter $\lambda_{wd}$ in eq. 4.

Figure 1 illustrates the accuracy and cross-entropy loss on the adversarial examples built for the held-out validation set for three choices[1] of $\lambda_{wd}$ throughout training. As seen in Figure 1(a), for small $\lambda_{wd}$ choices, the robust validation accuracy does not monotonically increase towards the end of training. The Non-monotonicity behavior, which is related to robust overfitting, is even more pronounced if we look at the robustness loss computed on the held-out validation (Figure 1(b)). Note that this behavior is still evident even if we look at the best hyper-parameter value according to the validation set ($\lambda_{wd}^* = 0.00089$).

Various methods have been proposed to rectify robust overfitting, including early stopping (Rice et al., 2020), use of extra unlabeled data (Carmon et al., 2019), synthesized images (Gowal et al., 2020), pre-training (Hendrycks et al., 2019a), use of data augmentations (Rebuffi et al., 2021), and stochastic weight averaging (Izmailov et al., 2018).

In Fig. 1, we observe that simply having smaller weight-norms (by increasing $\lambda_{wd}$) could reduce this non-monotonic behavior on the validation set adversarial examples. Although, this comes at the cost of larger cross-entropy loss on the training set adversarial examples, as shown in Figure 1(c). Even though the overall loss function from eq. 4 is a minimization problem, the terms in the loss function implicitly have conflicting objectives: During the training process, when the cross-entropy term holds dominance, effectively reducing the weight norm becomes challenging, resulting in non-monotonic behavior of robust validation metrics towards the later stages of training. Conversely, when the weight-norm term takes precedence, the cross-entropy objective encounters difficulties in achieving significant reductions. In the next section, we introduce *Adaptive Weight Decay*, which explicitly strikes a balance between these two terms during training.

## 2.2 Adaptive Weight Decay

Inspired by the findings in 2.1.2, we propose **A**daptive **W**eight **D**ecay (AWD). The goal of AWD is to maintain a balance between weight decay and cross-entropy updates during training in order to guide

---

[1]Figure 2 captures the complete set of $\lambda_{wd}$ values we tested.

the optimization to a solution which satisfies both objectives more effectively. To derive AWD, we study one gradient descent step for updating the parameter $w$ at step $t + 1$ from its value at step $t$:

$$w_{t+1} = w_t - \nabla w_t \cdot lr - w_t \cdot \lambda_{wd} \cdot lr, \tag{5}$$

where $\nabla w_t$ is the gradient computed from the cross-entropy objective, and $w_t \cdot \lambda_{wd}$ is the gradient computed from the weight decay term from eq. 3. We define $\lambda_{awd}$ as a metric that keeps track of the ratio of the magnituedes coming from each objective:

$$\lambda_{awd(t)} = \frac{\|\lambda_{wd} w_t\|}{\|\nabla w_t\|}, \tag{6}$$

To keep a balance between the two objectives, we aim to keep this ratio constant during training. AWD is a simple yet effective way of maintaining this balance. Adaptive weight decay shares similarities with non-adaptive (traditional) weight decay, with the only distinction being that the hyper-parameter $\lambda_{wd}$ is not fixed throughout training. Instead, $\lambda_{wd}$ dynamically changes in each iteration to ensure $\lambda_{awd(t)} \approx \lambda_{awd(t-1)} \approx \lambda_{awd}$. To keep this ratio constant at every step $t$, we can rewrite the $\lambda_{awd}$ equation (eq. 6) as:

$$\lambda_{wd(t)} = \frac{\lambda_{awd} \cdot \|\nabla w_t\|}{\|w_t\|}, \tag{7}$$

Eq. 7 allows us to have a different weight decay hyperparameter value ($\lambda_{wd}$) for every optimization iteration $t$, which keeps the gradients received from the cross entropy and weight decay balanced throughout the optimization. Note that weight decay penalty $\lambda_t$ can be computed on the fly with almost no computational overhead during the training. Using the exponential weighted average $\bar{\lambda}_t = 0.1 \times \bar{\lambda}_{t-1} + 0.9 \times \lambda_t$, we could make $\lambda_t$ more stable (Algorithm 1).

---

**Algorithm 1** Adaptive Weight Decay

1: **Input:** $\lambda_{awd} > 0$
2: $\bar{\lambda} \leftarrow 0$
3: **for** $(x, y) \in loader$ **do**
4:     $p \leftarrow model(x)$ ▷ Get models prediction.
5:     $main \leftarrow CrossEntropy(p, y)$ ▷ Compute CrossEntropy.
6:     $\nabla w \leftarrow backward(main)$ ▷ Compute the gradients of main loss w.r.t weights.
7:     $\lambda \leftarrow \frac{\|\nabla w\| \lambda_{awd}}{\|w\|}$ ▷ Compute iteration's weight decay hyperparameter.
8:     $\bar{\lambda} \leftarrow 0.1 \times \bar{\lambda} + 0.9 \times stop\_gradient(\lambda)$ ▷ Compute the weighted average as a scalar.
9:     $w \leftarrow w - lr(\nabla w + \bar{\lambda} \times w)$ ▷ Update Network's parameters.
10: **end for**

---

### 2.2.1 Differences between Adaptive and Non-Adaptive Weight Decay

To study the differences between adaptive and non-adaptive weight decay and to build intuition, we can plug in $\lambda_t$ of the adaptive method (eq. 7) directly into the equation for traditional weight decay (eq. 3) and derive the total loss based on Adaptive Weight Decay:

$$Loss_{w_t}(x, y) = Xent(f(x, w_t), y) + \frac{\lambda_{awd} \cdot \|\nabla w_t\| \|w_t\|}{2}, \tag{8}$$

Please note that directly solving eq. 8 will invoke the computation of second-order derivatives since $\lambda_t$ is computed using the first-order derivatives. However, as stated in Alg. 1, we convert the $\lambda_t$ into a non-tensor scalar to save computation and avoid second-order derivatives. We treat $\|\nabla w_t\|$ in eq. 8 as a constant and do not allow gradients to back-propagate through it. As a result, adaptive weight decay has negligible computation overhead compared to traditional non-adaptive weight decay.

By comparing the weight decay term in the adaptive weight decay loss (eq. 8): $\frac{\lambda_{awd}}{2}\|w\|\|\nabla w\|$ with that of the traditional weight decay loss (eq. 3): $\frac{\lambda_{wd}}{2}\|w\|^2$, we can build intuition on some of the differences between the two. For example, the non-adaptive weight decay regularization term approaches zero only when the weight norms are close to zero, whereas, in AWD, it also happens

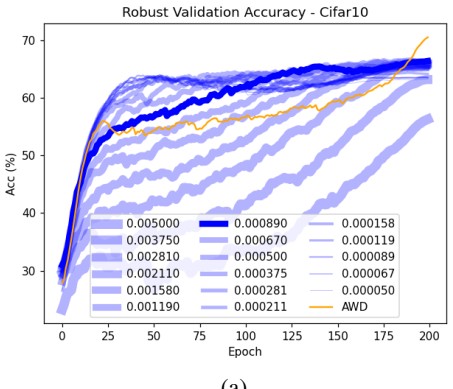
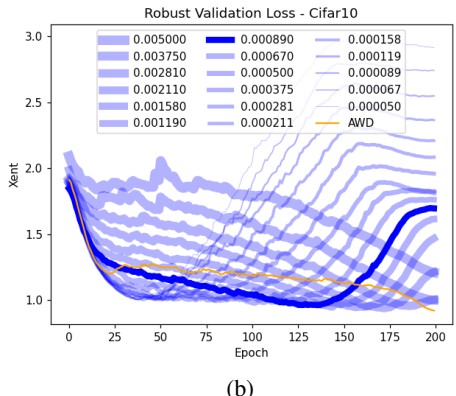

(a)                                   (b)

Figure 2: Robust accuracy (a) and loss (b) on CIFAR-10 validation subset. Both figures highlight the best performing hyper-parameter for non-adaptive weight decay $\lambda_{wd} = 0.00089$ with sharp strokes. As it can be seen, lower values of $\lambda_{wd}$ cause robust overfitting, while high values of it prevent network from fitting entirely. However, training with adaptive weight decay prevents overfitting and achieves highest performance in robustness.

when the cross-entropy gradients are close to zero. Consequently, AWD prevents over-optimization of weight norms in flat minima, allowing for more (relative) weight to be given to the cross-entropy objective. Additionally, AWD penalizes weight norms more when the gradient of cross-entropy is large, preventing it from falling into steep local minima and hence overfitting early in training.

We verify our intuition of AWD being capable of reducing robust overfitting in practice by replacing the non-adaptive weight decay with AWD and monitoring the same two metrics from 2.1.2. The results for a good choice of the AWD hyper-parameter ($\lambda_{awd}$) and various choices of non-adaptive weight decay ($\lambda_{wd}$) hyper-parameter are summarized in Figure 2 [2].

### 2.2.2   Related works to Adaptive Weight Decay

The most related studies to AWD are *AdaDecay* (Nakamura & Hong, 2019) and *LARS* (You et al., 2017). AdaDecay changes the weight decay hyper-parameter adaptively for each individual parameter, as opposed to ours which we tune the hyper-parameter for the entire network. LARS is a common optimizer when using large batch sizes which adaptively changes the learning rate for each layer. We evaluate these relevant methods in the context of improving adversarial robustness and experimentally compare with AWD in Table 2 and Appendix D [3].

### 2.3   Experimental Robustness results for Adaptive Weight Decay

AWD can help improve the robustness on various datasets which suffer from robust overfitting. To illustrate this, we focus on six datasets: SVHN, FashionMNIST, Flowers, CIFAR-10, CIFAR-100, and Tiny ImageNet. Tiny ImageNet is a subset of ImageNet, consisting of 200 classes and images of size $64 \times 64 \times 3$. For all experiments, we use the widely accepted 7-step PGD adversarial training to solve eq. 4 (Madry et al., 2017) while keeping 10% of the examples from the training set as held-out validation set for the purpose of early stopping. For early stopping, we select the checkpoint with the highest $\ell_\infty = 8$ robustness accuracy measured by a 3-step PGD attack on the held-out validation set. For CIFAR10, CIFAR100, and Tiny ImageNet experiments, we use a WideResNet 28-10 architecture, and for SVHN, FashionMNIST, and Flowers, we use a ResNet18 architecture. Other details about the experimental setup can be found in Appendix A.1. For all experiments, we tune the conventional non-adaptive weight decay parameter ($\lambda_{wd}$) for improving robustness generalization and compare that to tuning the $\lambda_{awd}$ hyper-parameter for adaptive weight decay. To ensure that we search for enough values for $\lambda_{wd}$, we use up to twice as many values for $\lambda_{wd}$ compared to $\lambda_{awd}$.

Figure 3 plots the robustness accuracy measured by applying AutoAttack (Croce & Hein, 2020b) on the test examples for the CIFAR-10, CIFAR-100, and Tiny ImageNet datasets, respectively. We

---

[2]See Appendix C.4 for similar analysis on other datasets.

[3]Due to space limitations we defer detailed discussions and comparisons to Appendix D.

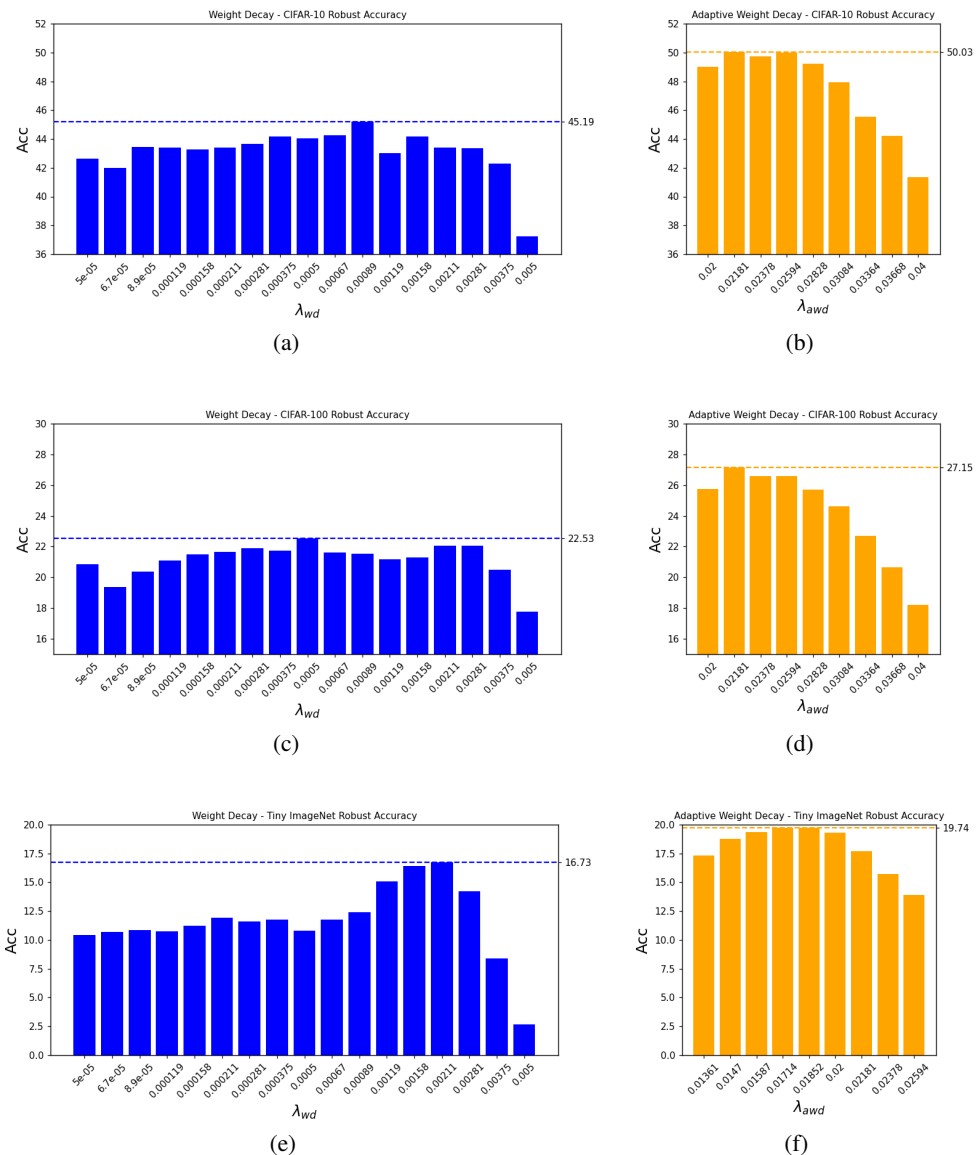

Figure 3: $\ell_\infty = 8$ robust accuracy on the test set of adversarially trained WideResNet28-10 networks on CIFAR-10, CIFAR-100, and Tiny ImageNet (a, c, e) using different choices for the hyper-parameters of non-adaptive weight decay ($\lambda_{wd}$), and (b, d, f) different choices of the hyper-parameter for adaptive weight decay ($\lambda_{awd}$).

observe that training with adaptive weight decay improves the robustness by a margin of 4.84% on CIFAR-10, 5.08% on CIFAR-100, and 3.01% on Tiny ImageNet, compared to the non-adaptive counterpart. These margins translate to a relative improvement of 10.7%, 20.5%, and 18.0%, on CIFAR-10, CIFAR-100, and Tiny ImageNet, respectively.

Increasing robustness often comes at the cost of drops in clean accuracy (Zhang et al., 2019). This observation could be attributed, at least in part, to the phenomenon that certain $\ell_p$-norm bounded adversarial examples bear a closer resemblance to the network's predicted class than their original class (Sharif et al., 2018). An active area of research seeks a better trade-off between robustness and natural accuracy by finding other points on the Pareto-optimal curve of robustness and accuracy. For example, (Balaji et al., 2019) use instance-specific perturbation budgets during training. Interestingly, when comparing the most robust network trained with non-adaptive weight decay ($\lambda_{wd}$) to that

| Method | Dataset | Opt | $\|W\|_2$ | Nat Acc | AutoAtt | $Xent + \frac{\lambda_{wd}^* \cdot \|W\|_2^2}{2}$ |
|---|---|---|---|---|---|---|
| $\lambda_{wd} = 0.00089$ | CIFAR-10 | SGD | 35.58 | 84.31 | 45.19 | 0.58 |
| $\lambda_{awd} = 0.022$ | | SGD | **7.11** | **87.08** | **50.03** | **0.08** |
| $\lambda_{wd} = 0.0005$ | CIFAR-100 | SGD | 51.32 | 60.15 | 22.53 | **0.67** |
| $\lambda_{awd} = 0.022$ | | SGD | **13.41** | **61.39** | **27.15** | 1.51 |
| $\lambda_{wd} = 0.00211$ | Tiny ImgNet | SGD | 25.62 | 47.87 | 16.73 | 3.56 |
| $\lambda_{awd} = 0.01714$ | | SGD | **15.01** | **48.46** | **19.74** | **2.80** |
| $\lambda_{wd} = 5e-7$ | SVHN | SGD | 102.11 | 92.04 | 44.16 | **1.02** |
| $\lambda_{awd} = 0.02378$ | | SGD | **5.39** | **93.04** | **47.10** | 1.15 |
| $\lambda_{wd} = 0.00089$ | FashionMNIST | SGD | 14.39 | 83.96 | 78.73 | 0.51 |
| $\lambda_{awd} = 0.01414$ | | SGD | **9.05** | **85.42** | **79.24** | **0.44** |
| $\lambda_{wd} = 0.005$ | Flowers | SGD | 19.94 | **90.98** | 32.35 | 1.72 |
| $\lambda_{awd} = 0.06727$ | | SGD | **13.87** | 90.39 | **39.22** | **1.42** |

Table 1: Adversarial robustness of PGD-7 adversarially trained networks using adaptive and non-adaptive weight decay. Table summarizes the best performing hyper-parameter for each method on each dataset. Not only the adaptive method outperforms the non-adaptive method in terms of robust accuracy, it is also superior in terms of the natural accuracy. Models trained with AWD have considerably smaller weight-norms. In the last column, we report the total loss value of the non-adaptive weight decay for the best tuned $\lambda$ for that dataset, found by grid search for each dataset. Interestingly, when we measure the non-adaptive total loss (eq. 4) on the training set, we observe that networks trained with the adaptive method often have smaller non-adaptive training loss even though in AWD we have not optimized that loss directly.

trained with AWD ($\lambda_{awd}$), we notice that those trained with the adaptive method have higher clean accuracy across various datasets (Table. 1).

In addition, we observe comparatively smaller weight-norms for models trained with adaptive weight decay, which might contribute to their better generalization and lesser robust overfitting. For the SVHN dataset, the bes model trained with AWD has $\approx 20x$ smaller weight-norm compared to the best model trained with traditional weight-decay. Networks which have such small weight-norms that maintain good performance on validation data is difficult to achieve with traditional weight-decay as previously illustrated in sec. 2.1.2. Perhaps most interestingly, when we compute the value of non-adaptive weight decay loss for AWD trained models, we observe that they are even sometimes superior in terms of that objective as it can be seen in the last column of Table 1. This behavior could imply that the AWD reformulation is better in terms of optimization and can more easily find a balance and simultaneously decrease both objective terms in eq. 4 and is aligned with the intuitive explanation in sec. 2.2.1.

The previously shown results suggest an excellent potential for adversarial training with AWD. To further study this potential, we only substituted the Momentum-SGD optimizer used in all previous experiments with the ASAM optimizer (Kwon et al., 2021), and used the same hyperparameters used in previous experiments for comparison with advanced adversarial robustness algorithms. To the best of our knowledge, and according to the RobustBench (Croce et al., 2020), the state-of-the-art $\ell_\infty = 8.0$ defense for CIFAR-100 without extra synthesized or captured data using WRN28-10 achieves 29.80% robust accuracy (Rebuffi et al., 2021). We achieve *29.54%* robust accuracy, which is comparable to these advanced algorithms even though our approach is a *simple modification to weight decay*. This is while our proposed method achieves 63.93% natural accuracy which is $\approx 1\%$ higher. See Table 2 for more details. In addition to comparing with the SOTA method on WRN28-10, we compare with a large suite of strong robustness methods which report their performances on WRN32-10 in Table 2. In addition to these two architectures, in Appendix C.1, we test other architectures to demonstrate the robustness of the proposed method to various architectural choices. For ablations demonstrating the robustness of AWD to various other parameter choices for adversarial training such as number of epochs, adversarial budget ($\epsilon$), please refer to Appendix C.3 and Appendix C.2, respectively.

Adaptive Weight Decay (AWD) can help improve the robustness over traditional weight decay on many datasets as summarized before in Table 1. In Table 2 we demonstrated that AWD when

| Method | WRN | Aug | Epo | ASAM | TR | SWA | Nat | AA |
|---|---|---|---|---|---|---|---|---|
| $\lambda_{AdaDecay} = 0.002$* | 28-10 | P&C | 200 | - | - | - | 57.17 | 24.18 |
| (Rebuffi et al., 2021) | 28-10 | CutMix | 400 | - | ✓ | ✓ | 62.97 | **29.80** |
| (Rebuffi et al., 2021) | 28-10 | P&C | 400 | - | ✓ | ✓ | 59.06 | 28.75 |
| $\lambda_{wd} = 0.0005$ | 28-10 | P&C | 200 | - | - | - | 60.15 | 22.53 |
| $\lambda_{wd} = 0.0005$ + ASAM | 28-10 | P&C | 100 | ✓ | - | - | 58.09 | 22.55 |
| $\lambda_{wd} = 0.00281$* + ASAM | 28-10 | P&C | 100 | ✓ | - | - | 62.24 | 26.38 |
| $\boldsymbol{\lambda_{awd} = 0.022}$ | 28-10 | P&C | 200 | - | - | - | 61.39 | 27.15 |
| $\boldsymbol{\lambda_{awd} = 0.022}$ + ASAM | 28-10 | P&C | 100 | ✓ | - | - | **63.93** | 29.54 |
| AT (Madry et al., 2017) | 32-10 | P&C | 100† | - | - | - | 60.13 | 24.76 |
| TRADES (Zhang et al., 2019) | 32-10 | P&C | 100† | - | ✓ | - | 60.73 | 24.90 |
| MART (Wang et al., 2020) | 32-10 | P&C | 100† | - | - | - | 54.08 | 25.30 |
| FAT (Zhang et al., 2020a) | 32-10 | P&C | 100† | - | - | - | **66.74** | 20.88 |
| AWP (Wu et al., 2020) | 32-10 | P&C | 100† | - | - | - | 55.16 | 25.16 |
| GAIRAT (Zhang et al., 2020b) | 32-10 | P&C | 100† | - | - | - | 58.43 | 17.54 |
| MAIL-AT (Liu et al., 2021) | 32-10 | P&C | 100† | - | - | - | 60.74 | 22.44 |
| MAIL-TR (Liu et al., 2021) | 32-10 | P&C | 100† | - | ✓ | - | 60.13 | 24.80 |
| $\boldsymbol{\lambda_{awd} = 0.022}$ + ASAM | 32-10 | P&C | 100 | ✓ | - | - | 64.49 | **29.70** |

Table 2: CIFAR-100 adversarial robustness performance of various strong methods. Adaptive weight decay with ASAM optimizer outperforms many strong baselines. For experiments marked with * we do another round of hyper-parameter search. $\lambda_{AdaDecay}$ indicates using the work from Nakamura & Hong (2019). The columns represent the method, depth and width of the WideResNets used, augmentation, number of epochs, whether ASAM, TRADES (Zhang et al., 2019), and Stochastic Weight Averaging (Izmailov et al., 2018), were used in the training, followed by the natural accuracy and adversarial accuracy using AutoAttack. In the augmentation column, P&C is short for Pad and Crop. The experiments with † are based on results from (Liu et al., 2021) which use a custom choice of parameters to alleviate robust overfitting. We also experimented with methods related to AWD such as LARS. We observed no improvement, so we do not report the results here. More details can be found in Appendix D.2.

combined with advanced optimization methods such as ASAM can result in models which have good natural and robust accuracies when compared with advanced methods on the CIFAR-100 dataset. Table 3 compares AWD+ASAM with various advanced methods on the CIFAR-10 dataset. The hyper-parameters used in this experiment are similar to those used before for the CIFAR-100 dataset. As it can be seen, despite it's simplicity, AWD depicts improvements over very strong baselines on two extensively studied datasets of the adversarial machine learning domain [4].

| Method | WRN | Aug | Epo | CIFAR-10 | |
|---|---|---|---|---|---|
| | | | | Nat | AA |
| AT (Madry et al., 2017) | 32-10 | P&C | 100† | 87.80 | 48.46 |
| TRADES (Zhang et al., 2019) | 32-10 | P&C | 100† | 86.36 | 53.40 |
| MART (Wang et al., 2020) | 32-10 | P&C | 100† | 84.76 | 51.40 |
| FAT (Zhang et al., 2020a) | 32-10 | P&C | 100† | **89.70** | 47.48 |
| AWP (Wu et al., 2020) | 32-10 | P&C | 100† | 57.55 | 53.08 |
| GAIRAT (Zhang et al., 2020b) | 32-10 | P&C | 100† | 86.30 | 40.30 |
| MAIL-AT (Liu et al., 2021) | 32-10 | P&C | 100† | 84.83 | 47.10 |
| MAIL-TR (Liu et al., 2021) | 32-10 | P&C | 100† | 84.00 | 53.90 |
| $\boldsymbol{\lambda_{awd} = 0.022}$ + ASAM | 32-10 | P&C | 100 | 88.55 | **54.04** |

Table 3: WRN32-10 models CIFAR-10 models trained with AWD when using the ASAM optimizer are more robust than models trained with various sophisticated algorithms from the literature.

---

[4]ImageNet robustness results can be seen in Appendix C.11

# 3 Additional Properties of Adaptive Weight Decay

Due to the properties mentioned before, such as reducing overfitting and resulting in networks with smaller weight-norms, Adaptive Weight Decay (AWD) can be seen as a good choice for other applications which can benefit from robustness. In particular, below in 3.1 we study the effect of adaptive weight decay in the noisy label setting. More specifically, we show roughly 4% accuracy improvement on CIFAR-100 and 2% on CIFAR-10 for training on the 20% symmetry label flipping setting (Bartlett et al., 2006). In addition, in the Appendix, we show the potential of AWD for reducing sensitivity to sub-optimal learning rates. Also, we show that networks which are naturally trained achieving roughly similar accuracy, once trained with adaptive weight decay, tend to have lower weight-norms. This phenomenon can have exciting implications for pruning networks (LeCun et al., 1989; Hassibi & Stork, 1992).

## 3.1 Robustness to Noisy Labels

Popular vision datasets, such as MNIST (LeCun & Cortes, 2010), CIFAR (Krizhevsky et al., 2009), and ImageNet (Deng et al., 2009), contain some amount of label noise (Yun et al., 2021; Zhang, 2017). While some studies provide methods for identifying and correcting such errors (Yun et al., 2021; Müller & Markert, 2019; Al-Rawi & Karatzas, 2018; Kuriyama, 2020), others provide training algorithms that avoid over-fitting, or even better, avoid fitting the incorrectly labeled examples entirely (Jiang et al., 2018; Song et al., 2019; Jiang et al., 2020).

In this section, we perform a preliminary investigation of adaptive weight decay's resistance to fitting training data with label noise. Following previous studies, we use symmetry label flipping (Bartlett et al., 2006) to create noisy data for CIFAR-10 and CIFAR-100 and use ResNet34 as the backbone. Other experimental setup details can be found in Appendix A.2. Similar to the previous section, we test different hyper-parameters for adaptive and non-adaptive weight decay. To ease comparison in this setting, we train two networks for each hyper-parameter: 1- with a certain degree of label noise and 2- with no noisy labels. We then report the accuracy on the clean label test set. The test accuracy on the second network – one which is trained with no label noise – is just the clean accuracy. Having the clean accuracy coupled with the accuracy after training on noisy data enables an easier understanding of the sensitivity of each training algorithm and choice of hyper-parameter to label noise. Figure 4 gathers the results of the noisy data experiment on CIFAR-100.

Figure 4 demonstrates that networks trained with adaptive weight decay exhibit a more modest decline in performance when label noise is present in the training set. For instance, Figure 4(a) shows that $\lambda_{awd} = 0.028$ for adaptive and $\lambda_{wd} = 0.0089$ for non-adaptive weight decay achieve roughly 70% accuracy when trained on clean data, while the adaptive version achieves 4% higher accuracy when trained on the noisy data. Appendix C.5 includes similar results for CIFAR-10.

Intuitively, based on eq. 7, in the later stages of training, when examples with label noise generate large gradients, adaptive weight decay intensifies the penalty for weight decay. This mechanism effectively prevents the model from fitting the noisy data by regularizing the gradients.

# 4 Conclusion

Regularization methods for a long time have aided deep neural networks in generalizing on data not seen during training. Due to their significant effects on the outcome, it is crucial to have the right amount of regularization and correctly tune training hyper-parameters. We propose Adaptive Weight Decay (AWD), which is a simple modification to weight decay – one of the most commonly employed regularization methods. In our study, we conduct a comprehensive comparison between AWD and non-adaptive weight decay in various settings, including adversarial robustness and training with noisy labels. Through rigorous experimentation, we demonstrate that AWD consistently yields enhanced robustness. By conducting experiments on diverse datasets and architectures, we provide empirical evidence to showcase the effectiveness of our approach in mitigating robust overfitting.

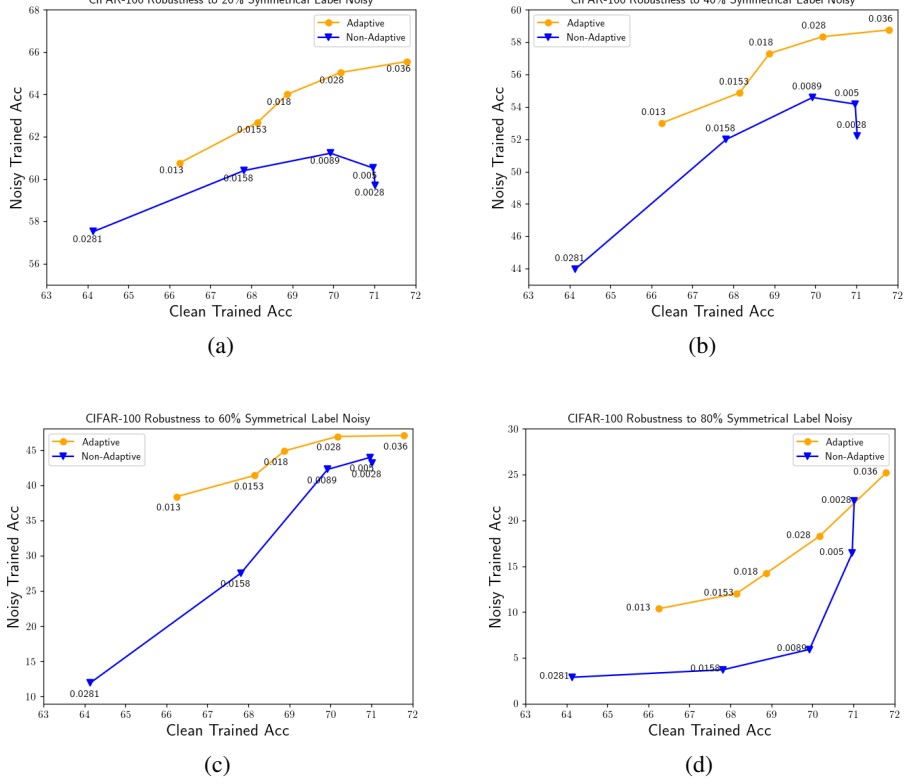

Figure 4: Comparison of similarly performing networks once trained on CIFAR-100 clean data, after training on 20% (a), 40% (b), 60% (c), and 80% (d). Networks trained with adaptive weight decay are less sensitive to label noise compared to ones trained with non-adaptive weight decay.

# 5   Acknowledgement

We extend our sincere appreciation to Zhile Ren for their invaluable support and perceptive contributions during the publication of this manuscript.

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
