# A  Experimental Setup

In this section, we include the experimental setup used to produce the experiments throughout this paper. We include all hyperparameters used for all experiments, unless explicitly mentioned otherwise. For all experiments, we use SGD optimizers with momentum $\mu = 0.9$. We use a Cosine learning rate schedule with no warm-up and with the value of $0$ for the final value. The weight decay and learning rate for experiments that have not been clearly specified are $\lambda_{wd} = 0.0005$ and $lr = 0.1$. We train all networks for 200 epochs with a batch-size of 128. For all experiments that use ASAM, we use the hyper-parameters the original paper suggests.

## A.1  Adversarial Training

We use a pre-activation WideResNet28 with a width of $10$. We use $\ell_\infty = 8/255$ PGD attack with the step size of $2/255$. We use 7 steps for creation of adversarial examples for training and use the minimum accuracy produced by AutoAttack for the test. We keep $10\%$ of training data for validation and use it to do early stopping. For $\lambda_{wd}$ of non-adaptive weight decay, we fit a geometric sequence of length 17 starting from $0.00005$ and ending at $0.005$. For the adaptive weight decay hyper parameter ($\lambda_{awd}$), we fit a geometric sequence of length 9 starting from $0.02$ and ending at $0.04$.

## A.2  Noisy Label Training

We use a ResNet34 as our architecture. For each setting we used 5 different hyperparameters for adaptive and non-adaptive weight decays. For CIFAR-10, we use $\lambda_{wd} \in \{0.0028, 0.005, 0.0089, 0.0158, 0.0281\}$ and $\lambda_{awd} \in \{0.036, 0.028, 0.018, 0.0153, 0.018\}$ and for CIFAR-100, we use $\lambda_{wd} \in \{0.0281, 9.9158, 0.0089, 0.005, 0.0028\}$ and $\lambda_{awd} \in \{0.013, 0.0153, 0.018, 0.028, 0.036\}$.

# B  Implementation

In this section, we discuss the details of implementation of adaptive weight decay. The method is not really susceptible to the exact implementation details discussed here, however, to be perfectly candid, we include all details here. First, let us assume that we desire to implement adaptive weight decay using $\lambda_{awd} = 0.016$ as the hyperparameter. We know that $\lambda_t = \frac{\|\nabla w_t\|0.016}{\|w_t\|}$. Please note that $\|\nabla w_t\|$ requires knowing the gradients of the loss w.r.t. the network's parameters. Meaning that to compute the $\lambda_t$ for every step, we have to call a backward pass on the actual parameters of the network. After this step, given the fact that we know both $\|w_t\|$ and $\|\nabla w_t\|$, we can compute $\lambda_t$.

# C  Extra Results

Here, we provide the extra results and figures not included in the body of the paper.

## C.1  Varying the architecture

To measure the dependency of AWD to architecture choices, we perform similar set of experiments to that in the main body but by varying the architecture. In particular, Table 4 summarizes the best performing hyper-parameter for Adaptive and Non-Adaptive Weight Decay methods on CIFAR-100 for various architectures. As it can be seen, AWD's considerable boost in robustness is not dependent on architecture. Not only adaptive weight decay outperforms the non-adaptive weight decay trained model in terms of robust accuracy, it is also superior in terms of the natural accuracy. Models trained with AWD have considerably smaller weight-norms to those trained with non-adaptive weight decay. In the last column, we report the value of the total loss of the non-adaptive weight decay for the best tuned non-adaptive ($\lambda_{wd}$) for that dataset which is found by doing a grid search. Interestingly, when we measure the non-adaptive total loss (eq. 4) on the training set, we observe that networks trained with the adaptive method often have smaller non-adaptive training loss even though in AWD we have not optimized that loss directly.

| Method | Arch | Opt | $\|W\|_2$ | Nat Acc | AutoAtt | $Xent + \frac{\lambda^*_{wd}\cdot\|W\|_2^2}{2}$ |
|---|---|---|---|---|---|---|
| $\lambda_{wd} = 0.00211$ | ResNet18 | SGD | 22.69 | 58.04 | 21.94 | 2.43 |
| $\lambda_{awd} = 0.02181$ | | SGD | **12.89** | **58.46** | **24.98** | **2.32** |
| $\lambda_{wd} = 0.00211$ | ResNet50 | SGD | 24.50 | 59.06 | 22.30 | 2.27 |
| $\lambda_{awd} = 0.02$ | | SGD | **13.87** | **60.60** | **26.73** | **2.02** |
| $\lambda_{wd} = 0.0005$ | WRN28-10 | SGD | 51.32 | 60.15 | 22.53 | **0.67** |
| $\lambda_{awd} = 0.022$ | | SGD | **13.41** | **61.39** | **27.15** | 1.51 |

Table 4: Adversarial robustness of PGD-7 adversarially trained CIFAR-100 networks using adaptive and non-adaptive weight decay on different architectures.

| $\epsilon$ | Data | Method | Nat | 20 | 40 | 60 | 80 | 100 | AA-SQ | AA-CE | AA-FAB | AA-T | AA |
|---|---|---|---|---|---|---|---|---|---|---|---|---|---|
| 2 | C10 | $\lambda_{wd}$ =0.00089 | 94.2 | 83.2 | 83.1 | 83.2 | 83.1 | 83.1 | 86.9 | 82.7 | 82.7 | 82.5 | 82.5 |
| 2 | C10 | $\lambda_{awd}$ =0.02181 | **94.3** | **83.6** | **83.6** | **83.6** | **83.6** | **83.6** | **87** | **83.2** | **83.1** | **83** | **83** |
| 2 | C100 | $\lambda_{wd}$ =0.00067 | 74.8 | 55.8 | 55.8 | 55.7 | 55.7 | 55.7 | 59.2 | 54.8 | 52.9 | 52.7 | 52.7 |
| 2 | C100 | $\lambda_{awd}$ =0.02181 | **75.2** | **56.7** | **56.7** | **56.7** | **56.6** | **56.7** | **59.6** | **56** | **53.7** | **53.4** | **53.4** |
| 4 | C10 | $\lambda_{wd}$ =0.00158 | 91.7 | 70.7 | 70.7 | 70.5 | 70.6 | 70.6 | 75.3 | 69.2 | 69.4 | 69 | 69 |
| 4 | C10 | $\lambda_{awd}$ =0.02181 | **92** | **73.1** | **73.1** | **73.1** | **73** | **73** | **77.8** | **72** | **71.7** | **71.3** | **71.3** |
| 4 | C100 | $\lambda_{wd}$ =0.00089 | 69.4 | 42 | 42 | 42 | 41.9 | 41.9 | 44.8 | 40.4 | 38.9 | 38.6 | 38.6 |
| 4 | C100 | $\lambda_{awd}$ =0.02181 | **71.5** | **46.8** | **46.7** | **46.7** | **46.7** | **46.7** | **48.3** | **45.2** | **41.2** | **40.8** | **40.8** |
| 6 | C10 | $\lambda_{wd}$ =0.00119 | 88.7 | 59 | 58.9 | 58.9 | 58.9 | 58.9 | 63.9 | 56 | 56.5 | 55.8 | 55.8 |
| 6 | C10 | $\lambda_{awd}$ =0.02181 | **90** | **62.4** | **62.3** | **62.4** | **62.3** | **62.3** | **67.3** | **60.3** | **60** | **59.5** | **59.5** |
| 6 | C100 | $\lambda_{wd}$ =0.00067 | 64.7 | 32.8 | 32.6 | 32.7 | 32.6 | 32.6 | 35 | 30.9 | 29.5 | 29.2 | 29.2 |
| 6 | C100 | $\lambda_{awd}$ =0.02181 | **66.3** | **39.6** | **39.5** | **39.5** | **39.5** | **39.5** | **39.8** | **37.7** | **33.4** | **33.1** | **33.1** |
| 8 | C10 | $\lambda_{wd}$ =0.00158 | 84 | 49.5 | 49.2 | 49.4 | 49.4 | 49.4 | 54.3 | 46.5 | 45.1 | 44.7 | 44.7 |
| 8 | C10 | $\lambda_{awd}$ =0.02181 | **87.3** | **53.9** | **53.8** | **53.7** | **53.8** | **53.8** | **58.1** | **51.4** | **50.1** | **49.6** | **49.6** |
| 8 | C100 | $\lambda_{wd}$ =0.00158 | 56.5 | 27.7 | 27.7 | 27.7 | 27.5 | 27.5 | 28.9 | 25.9 | 22.6 | 22.4 | 22.4 |
| 8 | C100 | $\lambda_{awd}$ =0.02181 | **61.6** | **33.1** | **33** | **33.1** | **33.1** | **33** | **32.5** | **31** | **26.7** | **26.4** | **26.4** |
| 16 | C10 | $\lambda_{wd}$ =0.00119 | 70.4 | 32 | 31.7 | 31.6 | 31.5 | 31.7 | 30.5 | 27.1 | 22.6 | 21.6 | 21.6 |
| 16 | C10 | $\lambda_{awd}$ =0.02181 | **71.9** | **34** | **33.8** | **33.7** | **33.7** | **33.7** | **33.2** | **29.6** | **26.1** | **25.3** | **25.3** |
| 16 | C100 | $\lambda_{wd}$ =0.00281 | 38.3 | 16.5 | 16.5 | 16.5 | 16.5 | 16.5 | 14.2 | 14.8 | 11.3 | 11 | 11 |
| 16 | C100 | $\lambda_{awd}$ =0.02181 | **41.5** | **19.8** | **19.7** | **19.6** | **19.5** | **19.5** | **17** | **17.7** | **13.9** | **13.4** | **13.4** |

Table 5: Adversarial robustness of PGD-7 adversarially trained WRN28-10 networks using adaptive and non-adaptive weight decay under various choices of $\epsilon$. Table summarizes the best performing hyper-parameter for non adaptive method, compared with the fixed hyper-parameter $\lambda_{awd} = 0.02181$. Not only the adaptive method outperforms the non-adaptive method in terms of robust accuracy, it is also superior in terms of the natural accuracy. Columns with numbers in the header, show the resulting robustness performance evaluated using multi-step PGD attacks. The final robust accuracy, which is the minimum accuracy over all attacks is gathered in the last column (AA.)

## C.2 Varying the Attack Budget

As discussed in Section 2.3, adaptive weight decay improves the performance of the network both in terms of robustness accuracy, as well as the natural accuracy, when the attack budget $\epsilon = 8$. To further show the applicability of the AWD method, we reproduced the CIFAR-10 and CIFAR-100 experiments with various $\epsilon$ budgets. We use the same budget $\epsilon$ for both training and evaluation of the network. Table 5 summarizes these results.

As it can be seen, regardless of the attack budget (i.e. $\epsilon$), the AWD trained models always outperform the non-adaptive counter parts, both in terms of natural and robustness accuracy.

| Epoch | Data | Method | Nat | 20 | 40 | 60 | 80 | 100 | AA-SQ | AA-CE | AA-FAB | AA-T | AA |
|---|---|---|---|---|---|---|---|---|---|---|---|---|---|
| 50 | C10 | $\lambda_{wd}$ =0.00158 | 86.5 | 52.6 | 52.4 | 52.4 | 52.4 | 52.5 | 56.7 | 49.5 | 48.5 | 48.0 | 48.0 |
| 50 | C10 | $\lambda_{awd}$ =0.02181 | **87.1** | **54.3** | **54.0** | **54.0** | **54.0** | **54.1** | **58.7** | **51.3** | **50.0** | **49.5** | **49.5** |
| 50 | C100 | $\lambda_{wd}$ =0.00211 | 59.5 | 29.8 | 29.7 | 29.7 | 29.7 | 29.6 | 31.1 | 28.0 | 25.3 | 25.1 | 25.1 |
| 50 | C100 | $\lambda_{awd}$ =0.02181 | **61.9** | **32.4** | **32.4** | **32.4** | **32.4** | **32.3** | **32.8** | **30.5** | **26.7** | **26.4** | **26.4** |
| 100 | C10 | $\lambda_{wd}$ =0.00211 | 85.8 | 50.8 | 50.6 | 50.5 | 50.6 | 50.6 | 55.4 | 47.7 | 47.1 | 46.6 | 46.6 |
| 100 | C10 | $\lambda_{awd}$ =0.02181 | **87.7** | **55.1** | **55.0** | **55.0** | **54.9** | **54.9** | **59.6** | **52.5** | **51.5** | **51.2** | **51.2** |
| 100 | C100 | $\lambda_{wd}$ =0.00281 | 58.2 | 28.2 | 28.1 | 28.0 | 28.1 | 28.1 | 29.4 | 26.3 | 23.7 | 23.4 | 23.4 |
| 100 | C100 | $\lambda_{awd}$ =0.02181 | **62.5** | **33.3** | **33.3** | **33.3** | **33.2** | **33.2** | **33.0** | **31.2** | **27.1** | **26.7** | **26.7** |
| 200 | C10 | $\lambda_{wd}$ =0.00158 | 84.1 | 50.3 | 50.1 | 50.1 | 50.0 | 50.0 | 54.8 | 47.4 | 46.2 | 45.7 | 45.7 |
| 200 | C10 | $\lambda_{awd}$ =0.02181 | **87.3** | **54.2** | **54.0** | **54.0** | **54.0** | **54.0** | **58.5** | **51.4** | **50.5** | **50.0** | **50.0** |
| 200 | C100 | $\lambda_{wd}$ =0.0005 | 60.5 | 25.2 | 25.2 | 25.1 | 25.0 | 25.1 | 27.0 | 23.1 | 22.4 | 22.2 | 22.2 |
| 200 | C100 | $\lambda_{awd}$ =0.02181 | **62.0** | **33.1** | **32.9** | **33.0** | **32.9** | **32.9** | **32.1** | **30.9** | **26.8** | **26.4** | **26.4** |
| 300 | C10 | $\lambda_{wd}$ =0.00089 | 86.2 | 48.9 | 48.8 | 48.6 | 48.6 | 48.6 | 52.8 | 44.9 | 45.2 | 44.5 | 44.5 |
| 300 | C10 | $\lambda_{awd}$ =0.02181 | **87.3** | **52.8** | **52.7** | **52.8** | **52.8** | **52.7** | **57.4** | **50.3** | **49.4** | **48.8** | **48.8** |
| 300 | C100 | $\lambda_{wd}$ =0.00028 | 59.5 | 25.6 | 25.5 | 25.5 | 25.5 | 25.5 | 27.0 | 23.6 | 22.6 | 22.3 | 22.3 |
| 300 | C100 | $\lambda_{awd}$ =0.02181 | **62.0** | **33.1** | **32.9** | **33.0** | **32.9** | **32.9** | **32.1** | **30.9** | **26.8** | **26.4** | **26.4** |

Table 6: Adversarial robustness of PGD-7 adversarially trained WRN28-10 networks using adaptive and non-adaptive weight decay. Table summarizes the best performing hyper-parameter for non adaptive method, compared with the fixed hyper-parameter $\lambda_{awd} = 0.022$. Not only the adaptive method outperforms the non-adaptive method in terms of robust accuracy, it is also superior in terms of the natural accuracy. Columns with numbers in the header, show the resulting robustness performance evaluated using multi-step PGD attacks.

## C.3 Varying the number of Epochs

In this section, we investigate AWD's performance and sensitivity to the length of training (number of epochs). To do so, we adversarially train WRN28-10 networks using PGD-7 with $\epsilon = 8$. Table 6 summarizes these results for the CIFAR-10 and CIFAR-100 datasets. As it can be seen, for various choices of training epochs, AWD outperfroms traditional weight decay both in natural accuracy and in robustness accuracy measured with AutoAttack (AA).

To further study the performance AWD in low epoch settings, we also reproduced the results of Table 6 with a WRN32-10 architecture and by varying the number of epochs. Table 7 summarizes these results. As illustrated, AWD's performance is not very sensitive to training time. While at 100 epochs, AWD's results (both nat and robustness) are comparable to models trained with 200 epochs, we found that even if we further reduce the epochs to 50, we do not see a big degradation of robust accuracy, although the natural accuracy degrades slightly.

## C.4 CIFAR-100 robustness and Adaptive Weight Decay

Figure 5 shows similar results to that of CIFAR-10 presented in Figure 1.

## C.5 CIFAR-10 robustness to noisy labels

The results of the experiments for training classifiers on the CIFAR-10 dataset with noisy labels can be seen in Figure 6 which yields similar conclusions to that of CIFAR-100.

## C.6 2D Grid search for best parameter values for ResNet32

The importance of the 2D grid search on learning-rate and weight decay hyper-parameters are not network dependent. And we can see how these values are tied together for ResNet32 in Figure 7.

| Epoch | Data | Method | Network | Nat | AA |
|---|---|---|---|---|---|
| 5 | C100 | $\lambda_{awd} = 0.022$ + ASAM | WRN32-10 | 26.79 | 11.03 |
| 10 | C100 | $\lambda_{awd} = 0.022$ + ASAM | WRN32-10 | 38.10 | 15.92 |
| 20 | C100 | $\lambda_{awd} = 0.022$ + ASAM | WRN32-10 | 51.27 | 21.67 |
| 30 | C100 | $\lambda_{awd} = 0.022$ + ASAM | WRN32-10 | 58.10 | 25.03 |
| 40 | C100 | $\lambda_{awd} = 0.022$ + ASAM | WRN32-10 | 62.01 | 27.51 |
| 50 | C100 | $\lambda_{awd} = 0.022$ + ASAM | WRN32-10 | 62.85 | 29.25 |
| 100 | C100 | $\lambda_{awd} = 0.022$ + ASAM | WRN32-10 | **64.49** | 29.70 |
| 150 | C100 | $\lambda_{awd} = 0.022$ + ASAM | WRN32-10 | 64.17 | **29.94** |
| 200 | C100 | $\lambda_{awd} = 0.022$ + ASAM | WRN32-10 | 64.37 | 29.55 |
| 250 | C100 | $\lambda_{awd} = 0.022$ + ASAM | WRN32-10 | 63.24 | 29.68 |
| 300 | C100 | $\lambda_{awd} = 0.022$ + ASAM | WRN32-10 | 63.35 | 29.28 |

Table 7: Adversarial robustness of PGD-7 adversarially trained WRN32-10 networks using adaptive weight decay with fixed hyper-parameter $\lambda_{awd} = 0.022$ and varying number of training epochs. The adaptive method has an acceptable performance in settings with low training epochs, even as low as 50 epochs.

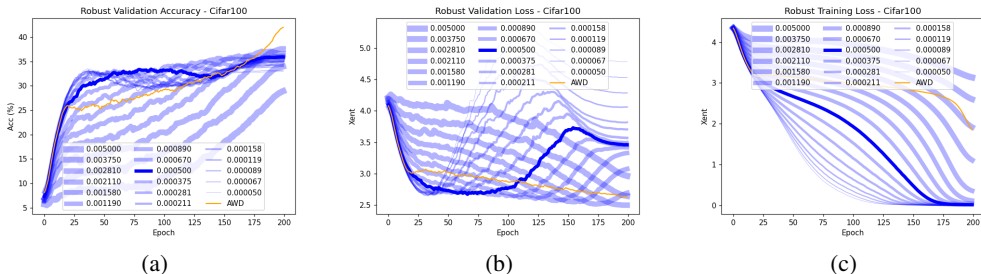

(a)          (b)          (c)

Figure 5: Robust validation accuracy (a), validation loss (b), and training loss (c) on CIFAR-100 held-out validation subset. $\lambda_{wd} = 0.0005$ is the best performing hyper-parameter we found by doing a grid-search. The thickness of the plot-lines correspond to the magnitude of the weight-norm penalties. As it can be seen by (a) and (b), networks trained by small values of $\lambda_{wd}$ suffer from robust-overfitting, while networks trained with larger values of $\lambda_{wd}$ do not suffer from robust overfitting but the larger $\lambda_{wd}$ further prevents the network from fitting the data (c) resulting in reduced overall robustness.

## C.7 Visualizing images from the CIFAR-100 training set where best AWD models do not fit.

In Figure 8, we visualize some of the 4.71% examples which belong to the CIFAR-100 training set that our AWD trained network doese not fit. Interestingly enough, there are many examples like 0-3, 10-14, 48, and 49 with overlapping classes with one object. There are many examples with wrong labels, such as 4-6, 43-45, 28-29, and 51-57. In many more examples, there are at least two objects in one image, such as 15-25, 28-33, 38-42, and 58-59. Unsurprisingly, the model would be better off not fitting such data, as it would only confuse the model to fit the data, which contradicts its already existing and correct conception of the object.

## C.8 Additional Robustness benefits

Throughout the 2D grid search experiments in section C.6, we observed that non-adaptive weight decay is sensitive to changes in the learning rate (LR). In this section we aim to study the sensitivity of the best hyper-parameter value for adaptive and non-adaptive weight decay to learning rate. In addition, models trained with adaptive weight decay tend to have smaller weight norms which could make them more suited for pruning. To test this intuition, we adopt a simple non-iterative $\ell_1$ pruning. To build confidence on robustness to LR and pruning, for the optimal choices of $\lambda_{wd} = 0.0005$ and the estimated $\lambda_{awd} = 0.016$ for WRN28-10, we train 5 networks per choice of learning rate. We prune each network to various degrees of sparsity. We then plot the average of all trials per parameter set for each of the methods. Figure 9 summarizes the clean accuracy without any pruning and the

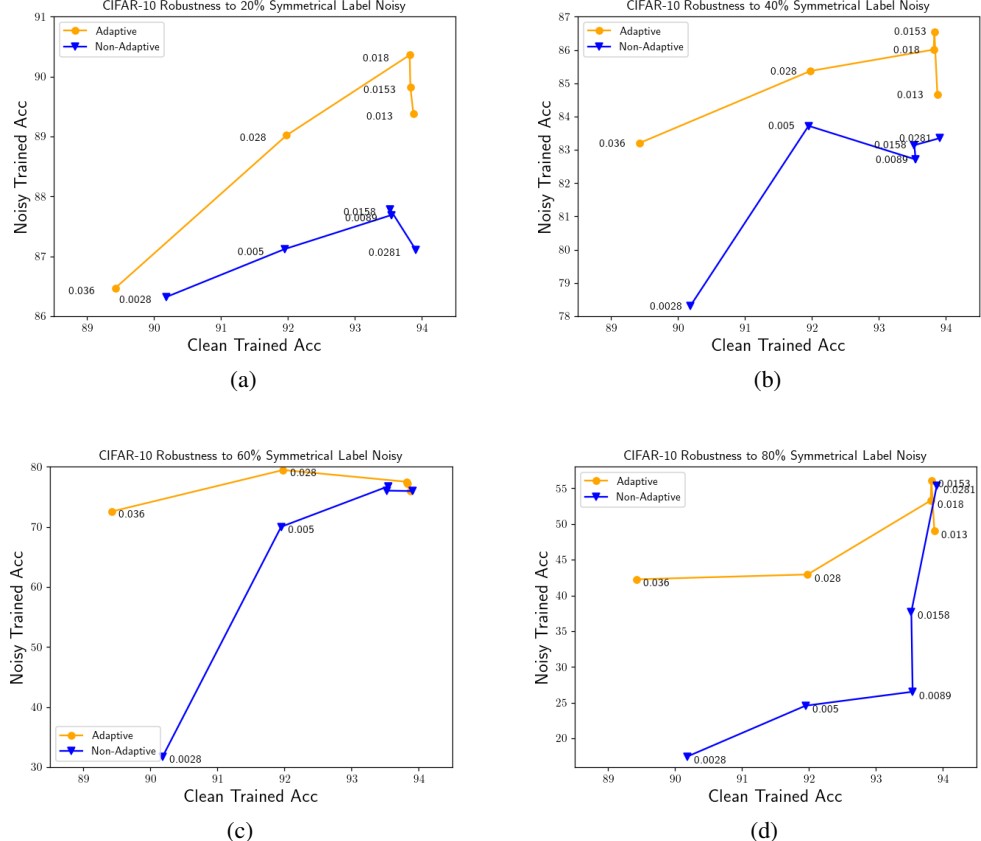

Figure 6: Comparison of similarly performing networks once trained on CIFAR-100 clean data, after training on 20% (a), 40% (b), 60% (c), and 80% (d) noisy data. Networks trained with adaptive weight decay outperform non-adaptive trained networks.

accuracy after 70% of the network is pruned. As it can be seen, for CIFAR-100, adaptive weight decay is both more robust to learning rate changes and also the result networks are less sensitive to parameter pruning. For more details and results on CIFAR-10, please see Appendix C.10 and C.12.

## C.9 Under-Fitting Data, A Desirable Property

Our experiments show that adaptive weight decay prevents fitting all the data in case of noisy label and adversarial training. Experimentally, we showed that adaptive weight decay contributes to this outcome more than non-adaptive weight decay. Interestingly, even in the case of natural training of even simple datasets such as CIFAR-100, networks trained with optimal adaptive weight decay still underfit the data. For instance, consider the following setup where we train a ResNet50, with ASAM (Kwon et al., 2021) minimizer with both adaptive and non-adaptive weight decay[5]. Table 8 shows the accuracy of the two experiments.

Intruigingly, without losing any performance, the model trained with adaptive weight decay avoids fitting an extra 3.28% of training data compared to the model trained with non-adaptive weight decay. We investigated more on what the 3.28% unfit data looks like [6]. Previously studies have discovered that some of the examples in the CIFAR-100 dataset have wrong labels (Zhang, 2017; Müller & Markert, 2019; Al-Rawi & Karatzas, 2018; Kuriyama, 2020). We found the 3.28% to be consistent with their findings and that many of the unfit data have noisy labels. We show some apparent noisy labeled examples in Figure 10.

---

[5]To find the best performing hyper-parameters for both settings, we do a 2D grid-search similar to Figure 7.
[6]See Appendix C.7 for images from the 4.71%.

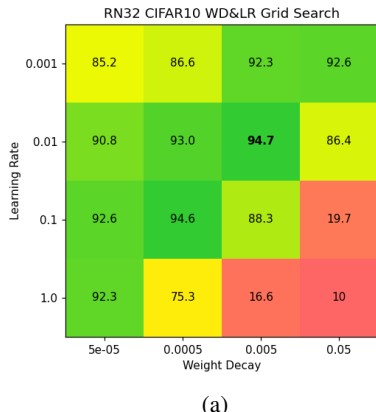

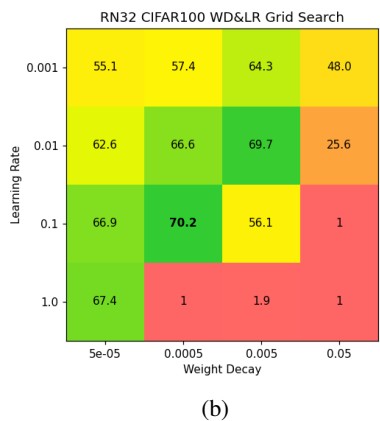

| | (a) | | | | (b) | |

Figure 7: Grid Search on different values of learning rate and weight decay on accuracy of ResNet32 on (a) CIFAR10 and (b) CIFAR100.

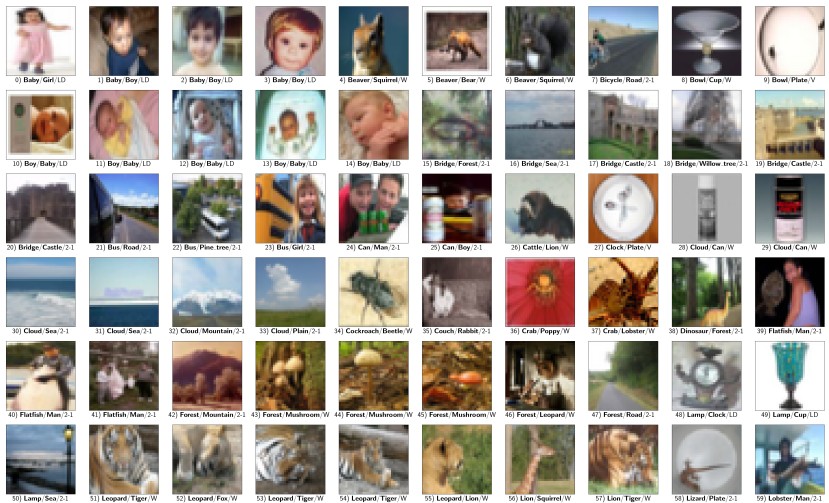

Figure 8: Examples from CIFAR-100 training dataset that have noisy labels. For every image we state [dataset label/ prediction of classifier trained with AWD / our category of noisy case]. We classify these noisy labels into several categories: **W**: Wrong Labels where the picture is clear enough to comprehend the correct label.; **2-1**: Two Objects from CIFAR-100 in one image, but only one label is given in the dataset; **LD**: Loosely Defined Classes where there is one object, but one object could be two classes at the same time. For instance, a baby girl is both a baby and a girl. **V**: Vague images where authors had a hard time identifying.

## C.10  Robustness to Sub-Optimal Learning Rate

In this section, we have a deeper look the performance of networks trained with sub-optimal learning rates. We observe that adaptive weight decay is more robust to changes in the learning rate in comparison to non-adaptive weight decay for both CIFAR-10 and CIFAR-100, as shown in Figure 11(a) and 11(b). The accuracy for $lr = 1.0$ for non-adaptive weight decay drops 69.67% on CIFAR-100 and 5.0% on CIFAR-10, compared to its adaptive weight decay counterparts.

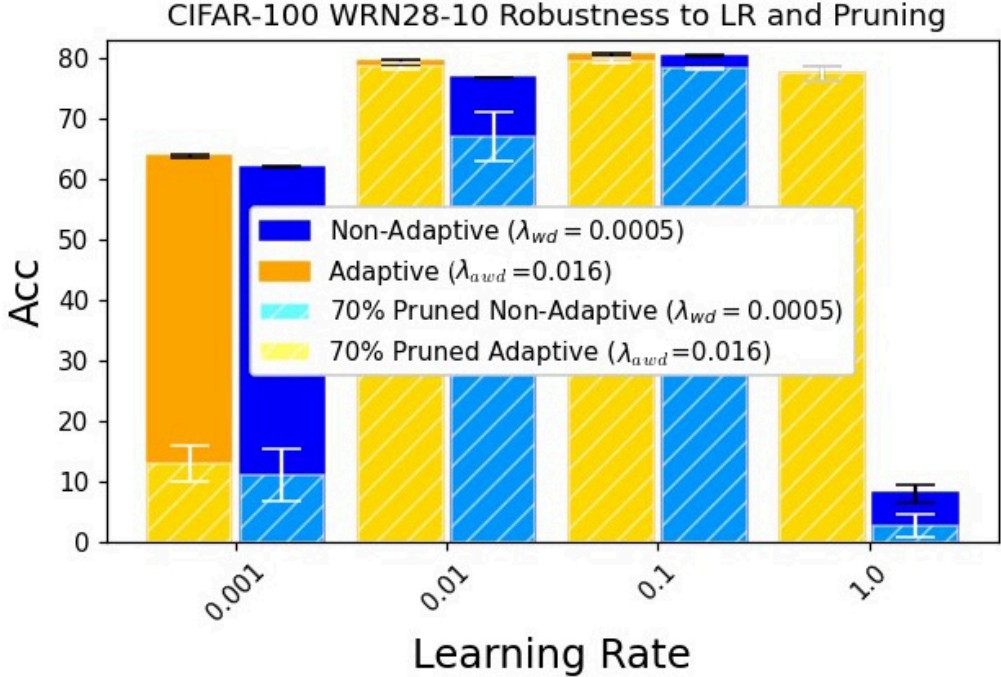

Figure 9: CIFAR-100 models trained with Adaptive Weight Decay (AWD) are less sensitive to learning rate. Also, due to the smaller weight norms of models trained with AWD, they seem like good candidates for pruning. Interestingly, when models are trained with smaller learning rates, they could be more sensitive to trivial pruning algorithms such as non-iterative (i.e., global) $\ell_1$ pruning. The results are average of 4 runs.

| Method | Learning Rate | $\|W\|_2$ | Test Acc(%) | Training Acc(%) |
|---|---|---|---|---|
| Adaptive $\lambda_{awd} = 0.022$ | 0.01 | 14.39 | 83.21 | 95.29 |
| Non-Adaptive $\lambda_{wd} = 0.005$ | 0.01 | 17.86 | 83.23 | 98.57 |

Table 8: Train and Test accuracy of adaptive and non-adaptive weight decay trained models. While the adaptive version fits 3.28% less training data, it still results in comparable test accuracy. Our hypothesis is that probably it avoids fitting the noisy labeled data.

The robustness to sub-optimal learning rates suggests that adaptive weight decay might be more suitable for applications where tuning for the optimal learning rate might be expensive or impractical. An example would be large language models such as GPT-3 or Megatron-Turing NLG, where even training the network once is expensive (Brown et al., 2020; Smith et al., 2022; Rasley et al., 2020). Another example would be neural architecture search, where one trains many networks (Tan & Le, 2019; Zhou et al., 2018; Real et al., 2019; Bergstra et al., 2013; Mendoza et al., 2016).

### C.11 Adaptive weight decay on ImageNet

In this section we illustrate that tuning adaptive weight decay can result in comparable performance to non-adaptive weight decay even in settings where training with non-adaptive weight decay does not suffer from over-fitting. For this purpose we perform free adversarial training (Shafahi et al., 2019) with $\epsilon = 4/255$ on ImageNet scale. We use a resnet-50 backbone. The parameters used in this setting are replay $m = 4$, batchsize of 512, and an initial learning rate of 0.1 which drops by a factor of 0.1 each $n/3$ epochs.

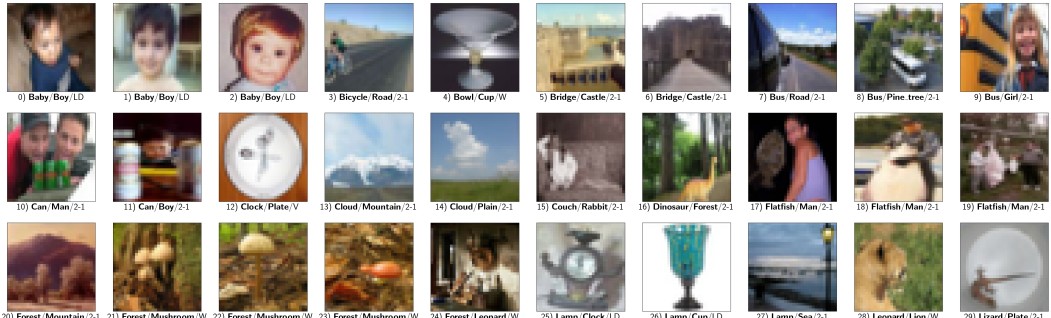

Figure 10: Examples from CIFAR-100 training dataset that have noisy labels. For every image we state [dataset label/ prediction of classifier trained with AWD / our category of noisy case]. We classify these noisy labels into several categories: **W**: Wrong Labels where the picture is clear enough to comprehend the correct label.; **2-1**: Two Objects from CIFAR-100 in one image, but only one label is given in the dataset; **LD**: Loosely Defined Classes where there is one object, but one object could be two classes at the same time. For instance, a baby girl is both a baby and a girl. **V**: Vague images where authors had a hard time identifying.

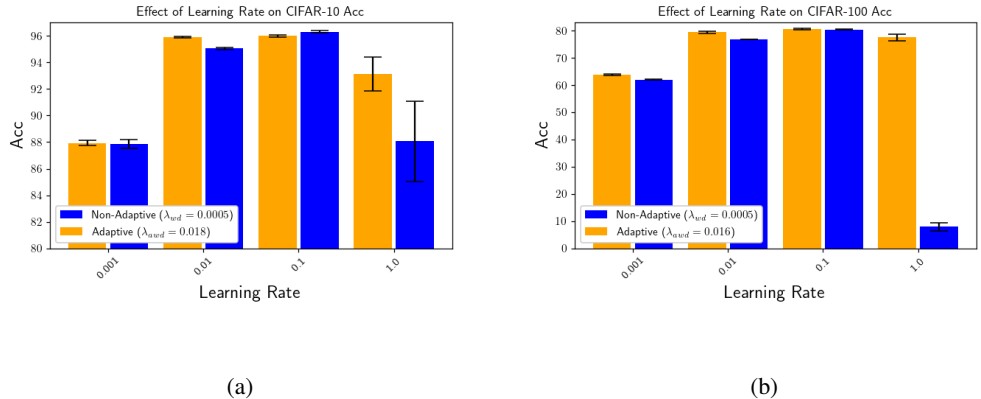

(a)                      (b)

Figure 11: Models trained with Adaptive Weight Decay are more robust to Learning rate. Results are an average over 5 trials.

Similar to the experiments in the main body, we perform a hyper-parameter search to find the best weight decay parameter and the best adaptive weight decay parameter and report both the robustness and clean accuracy of each of the trained models. The results are summarized in Table 9.

### C.12 Robustness to Parameter Pruning

As seen in previous sections, models trained with adaptive weight decay tend to have smaller weight norms. Smaller weight norms, can indicate that models trained with adaptive weight decay are less sensitive to parameter pruning. We adopt a simple non-iterative $\ell_1$ pruning to test our intuition. Table 10 which shows the accuracy of models trained with adaptive and non-adaptive weight decay after various percentages of the parameters have been pruned verifies our hypothesis. Adaptive weight decay trained models which are trained with various learning rates are more robust to pruning in comparison to models trained with non-adaptive weight decay.

| Method | Model | $\epsilon$ | robustness % | natural accuracy % |
|---|---|---|---|---|
| $\lambda_{wd} = 0.000004$ | Resnet-50 | 4 | 26.04 | 55.41 |
| $\lambda_{wd} = 0.000005$ | Resnet-50 | 4 | 26.71 | 54.95 |
| $\lambda_{wd} = 0.000006$ | Resnet-50 | 4 | 25.84 | 55.13 |
| $\lambda_{wd} = 0.000007$ | Resnet-50 | 4 | 25.71 | 54.20 |
| $\lambda_{awd} = 0.0006$ | Resnet-50 | 4 | 25.94 | 54.06 |
| $\lambda_{awd} = 0.0007$ | Resnet-50 | 4 | **26.84** | **56.31** |
| $\lambda_{awd} = 0.0008$ | Resnet-50 | 4 | 26.36 | 54.99 |
| $\lambda_{awd} = 0.0009$ | Resnet-50 | 4 | 26.39 | 55.69 |

Table 9: Robustness and clean accuracy of resnet-50 models trained with adversarial training for free $m = 4$ to be robust against attacks with robustness budget of $\epsilon = 4$.

| Method | Learning Rate | | | |
|---|---|---|---|---|
| | 0.001 | 0.01 | 0.1 | 1.0 |
| Adaptive $\lambda_{awd} = 0.016$ | **92.53** | **16.22** | **18.3** | 26.65 |
| Non-Adaptive $\lambda_{wd} = 0.0005$ | 116.98 | 27.01 | 24.06 | **12.16** |

Figure 12: Norm of the weights for networks trained with adaptive weight decay and non-adaptive weight decay.

# D   Related Work on Adaptive Weight Decay

As discussed before, AdaDecay Nakamura & Hong (2019) and Lars You et al. (2017) are the methods most related to ours. Here we discuss the major differences between our method and the related works.

## D.1   AdaDecay

The concept of Adaptive Weight Decay was first introduced by (Nakamura & Hong, 2019). Similar to our method, their method (AdaDecay) changes the weight decay's hyper-parameter at every iteration. Unlike our method, in one iteration, AdaDecay imposes a different penalty to each individual parameter, while our method penalizes all parameters with the same magnitude. For instance, let us consider the weight decay updates from eq 5 for both methods. For parameters $w$ at iteration $t$, the weight decay updates in AWD is $-\lambda_t w$, where $\lambda_t$ varies at every iteration. However, the AdaDecay updates $w_i$ is $-\lambda \theta_{t,i} w_i$, where $\lambda$ is constant at every iteration, instead, $0 \le \theta_{t,i} \le 2$ can vary for different parameters $i$ and for different iterations $t$. In other words, AdaDecay introduces $\theta_{t,i}$ for every single parameter of the network at every step to represent how strongly each parameter should be penalized. For instance, if $\forall t, i : \theta_{t,i} = 1$, then AdaDecay has the same effect as traditional non-adaptive weight decay method.

To further understand the differences between AdaDecay and AWD, we build intuition on the values $\theta_{t,i}$ could possibly take. For parameters in layer $L$, Ada decay defines $\theta_{t,i}$ as:

$$\theta_{t,i} = \frac{2}{1 + exp(-\alpha \nabla \bar{w}_{i,t})} \tag{9}$$

where $\nabla \bar{w}_{i,t}$ is the layerwise-normalized gradients. More precisely, $\nabla \bar{w}_{i,t} = \frac{\nabla w_{i,t} - \mu_L}{\sigma_L}$ where $\mu_L$ and $\sigma_L$ represent mean and standard deviation of gradients at layer $L$ respectively. Please note that $\mathbb{E}_{w_i \in L} \theta_{t,i} = 1$, due to the fact that $\theta_{t,i}$ is a sigmoid function applied to a distribution with mean of zero and standard deviation of one. In other words, AdaDecay on average does not change the $\lambda_{wd}$ hyper-parameter, while AWD does. In simpler words, AWD can increase the weight norm penalty (i.e. $\lambda_t$) indefinitely, while AdaDecay's penalty is bounded. For instance, assuming the most extreme case for AdaDecay where $\nabla \bar{w}_{i,t} = \infty$ for all $i$ and $t$, then $\theta_{t,i} = 2$. In other words, the effect of AdaDecay becomes at most twice as strong as non-adaptive weight decay, while our version does not have such upper-bounds.

| Dataset | LR | Method | Nat | 40% | 50% | 60% | 70% | 80% | 90% |
|---|---|---|---|---|---|---|---|---|---|
| C10 | 1 | $\lambda_{awd} = 0.018$ | **93.1 ± 1.3** | **93.1 ± 1.3** | **93.1 ± 1.3** | **93.1 ± 1.3** | **93.1 ± 1.3** | **93.1 ± 1.3** | **93.2 ± 1.3** |
| | | $\lambda_{wd} = 0.0005$ | 88.1 ± 3.0 | 88.0 ± 3.1 | 88.0 ± 3.1 | 88.0 ± 3.1 | 88.0 ± 3.1 | 88.0 ± 3.1 | 88.0 ± 3.1 |
| | 0.1 | $\lambda_{awd} = 0.018$ | 96.0 ± 0.1 | 91.6 ± 8.7 | 91.7 ± 8.6 | 91.7 ± 8.6 | 91.5 ± 9.0 | 90.5 ± 10.5 | 83.7 ± 17.9 |
| | | $\lambda_{wd} = 0.0005$ | **96.3 ± 0.1** | 93.6 ± 5.3 | 93.6 ± 5.3 | 93.6 ± 5.3 | 93.4 ± 5.5 | 92.6 ± 6.2 | 84.3 ± 13.6 |
| | 0.01 | $\lambda_{awd} = 0.018$ | **95.9 ± 0.1** | **95.9 ± 0.1** | **95.9 ± 0.1** | **95.9 ± 0.1** | **95.9 ± 0.1** | **95.8 ± 0.1** | **91.4 ± 1.4** |
| | | $\lambda_{wd} = 0.0005$ | 95.0 ± 0.1 | 95.0 ± 0.1 | 95.0 ± 0.1 | 94.5 ± 0.3 | 75.2 ± 12.9 | 19.3 ± 12.6 | 12.2 ± 5.0 |
| | 0.001 | $\lambda_{awd} = 0.018$ | 87.9 ± 0.2 | 82.1 ± 1.9 | 75.2 ± 4.2 | 64.6 ± 7.2 | 38.3 ± 8.4 | 18.8 ± 4.3 | 13.4 ± 1.9 |
| | | $\lambda_{wd} = 0.0005$ | 87.9 ± 0.4 | 80.5 ± 2.9 | 73.8 ± 4.8 | 56.4 ± 8.1 | 46.8 ± 16.9 | 21.0 ± 9.9 | 12.2 ± 2.5 |
| C100 | 1 | $\lambda_{awd} = 0.016$ | **77.6 ± 1.3** | **77.6 ± 1.3** | **77.6 ± 1.3** | **77.6 ± 1.3** | **77.6 ± 1.3** | **77.5 ± 1.2** | **75.3 ± 1.3** |
| | | $\lambda_{wd} = 0.0005$ | 7.9 ± 1.4 | 2.6 ± 1.8 | 2.6 ± 1.8 | 2.6 ± 1.8 | 2.6 ± 1.8 | 2.6 ± 1.8 | 2.6 ± 1.8 |
| | 0.1 | $\lambda_{awd} = 0.016$ | 80.7 ± 0.3 | 80.7 ± 0.3 | **80.7 ± 0.3** | 80.5 ± 0.2 | 79.6 ± 0.4 | 74.5 ± 0.9 | 29.2 ± 3.6 |
| | | $\lambda_{wd} = 0.0005$ | 80.5 ± 0.2 | 80.4 ± 0.1 | 80.2 ± 0.2 | 79.9 ± 0.2 | 78.4 ± 0.2 | 71.3 ± 1.2 | 23.9 ± 6.7 |
| | 0.01 | $\lambda_{awd} = 0.016$ | **79.5 ± 0.4** | **79.5 ± 0.3** | 79.4 ± 0.3 | 79.3 ± 0.4 | 78.7 ± 0.5 | 75.0 ± 0.8 | 28.9 ± 4.4 |
| | | $\lambda_{wd} = 0.0005$ | 76.9 ± 0.1 | 74.4 ± 3.6 | 74.1 ± 3.8 | 73.5 ± 4.2 | 67.2 ± 4.1 | 17.5 ± 6.9 | 1.5 ± 0.4 |
| | 0.001 | $\lambda_{awd} = 0.016$ | **63.8 ± 0.3** | **55.3 ± 1.8** | **47.4 ± 2.2** | **32.0 ± 2.1** | 12.9 ± 2.9 | 2.8 ± 1.4 | 1.1 ± 0.2 |
| | | $\lambda_{wd} = 0.0005$ | 62.1 ± 0.2 | 52.7 ± 2.0 | 41.7 ± 2.7 | 25.6 ± 2.4 | 11.0 ± 4.4 | 3.1 ± 0.9 | 1.2 ± 0.1 |

Table 10: Models trained with Adaptive Weight Decay are more robust to change in learning rate and pruning.

We also performed experimental evaluations on AdaDecay. The results are summerized in Table 2. We performed a grid-search on the hyper-parameter for AdaDecay and based on PGD-3 adversarial robustness on a held-out validation set, selected the best performing hyper-parameter for AdaDecay. As it can be seen in Table 2, AWD trained network outperforms the AdaDecay trained network by 3% accuracy.

### D.2 LARS

LARS You et al. (2017) is an optimizer that has been widely adopted for large-batch optimization. LARS achieves amazing performance by adaptively changing the learning rate for each layer. At first glance, the LARS and AWD seem very similar, where in fact, they are different and can even be applied simultaneously during training. The similarities raise from the fact that both methods use $\frac{\|w\|_2}{\|\nabla w\|_2}$ as a signal to adaptively change training hyper-parameters, while LARS changes the learning rate and AWD changes the weight decay hyper parameter.

To better understand the differences between the two methods, let us rewrite eq 5 for LARS. At step $t + 1$ for a network with only one layer with parameter $w$, we will have:

$$w_{t+1} = w_t - lr \times \nu \times \frac{\|w\|}{\|\nabla w\| + \lambda \|w\|}(w + \nabla w) \tag{10}$$

Note that in this setting, the ratio between the updates from the main loss (i.e., $\nabla w$) and weight decay(i.e., $w$) is untouched, while AWD enforces stronger regularization effects by breaking this constant ratio. Applying AWD we have:

$$w_{t+1} = w_t - lr \times (\nabla w + \lambda \frac{\|\nabla w\|}{\|w\|} w) \tag{11}$$

Also, note that since the two methods are independent of one another, they can be combined. So using AWD and LARS together, we will have:

$$w_{t+1} = w_t - lr \times \nu \times \frac{\|w\|}{\|\nabla w\|}(\lambda \frac{\|\nabla w\|}{\|w\|} w + \nabla w) \tag{12}$$

To experimentally study the differences between the two methods, we repeat the experiments done for Table 1 with and without the LARS optimizer on a ResNet18 architecture. For LARS hyper-

parameters, we use trust-coefficient=0.02, $\epsilon = 1e-8$, and we clip the gradients. Table 11 summarizes the results of these experiments.

| Method | Dataset | Opt | $\|W\|_2$ | Nat Acc | AutoAtt | $Xent + \frac{\lambda_{wd}^* \cdot \|W\|_2^2}{2}$ |
|---|---|---|---|---|---|---|
| $\lambda_{wd} = 0.00089$ | | SGD | 28.42 | 83.43 | 43.20 | 1.11 |
| $\lambda_{awd} = 0.01834$ | CIFAR-10 | SGD | **7.89** | **85.99** | **46.87** | **0.80** |
| $\lambda_{wd} = 0.00089$ | | LARS | 27.87 | 83.79 | 43.28 | 1.11 |
| $\lambda_{awd} = 0.02181$ | | LARS | **6.20** | **84.74** | **46.82** | **0.99** |
| $\lambda_{wd} = 0.0.00211$ | | SGD | 22.69 | 58.04 | 21.94 | 2.43 |
| $\lambda_{awd} = 0.02181$ | CIFAR-100 | SGD | **12.89** | **58.46** | **24.98** | **2.32** |
| $\lambda_{wd} = 0.00211$ | | LARS | 22.91 | 57.82 | 21.72 | 2.40 |
| $\lambda_{awd} = 0.02181$ | | LARS | **13.11** | **58.90** | **24.89** | **2.23** |

Table 11: Adversarial robustness of PGD-7 adversarially trained ResNet18 on CIFAR10 and CIFAR100 datasets. Similar to the Table 1, we used a grid search to find the best $\lambda_{wd}$ hyper-parameter for non-adaptive rows and $\lambda_{awd}$ for adaptive rows. The table compares the effect of AWD and LARS. As can be seen, the AWD trained models outperform the non-adaptive counter-parts in terms of robustness, natural accuracy, smaller weight norms, and smaller non-adaptive training loss. Interestingly enough, LARS does not have any major effects on any of the mentioned metrics, unless combined with AWD.

# E   Limitations

As discussed throughout the paper, AWD is most effective when overfitting happens. In other words, AWD would not be effective in settings that we underfit the training data such as ImageNet dataset. Needless to say, in such settings, AWD does not hurt the accuracy either. Section C.11 summarizes our ImageNet results for the adversarial robustness setting. As it can be seen in Table 9, we gain minor improvements in terms of adversarial robustness by adopting AWD.

Similar to the traditional weight decay, AWD requires hyper-parameter tuning. Our experiments show that AWD works best if its hyper-parameter is tuned per dataset and per network architecture.

# F   An Example for Convergence of Adaptive Weight Decay

As discussed previously in Section 2.2.1, we treat the weight decay hyper-parameter computed in each iteration as a non-tensor constant and we do not let the gradients to back-propagate through the computation of $\lambda_{wd(t)}$ in eq. 7. So in some sense, every optimization step of the adaptive weight decay is just an optimization step in the traditional non-adaptive weight decay, with the only difference that the weight decay hyper-parameter is being scaled. Consequently, in terms of convergence, it is not unlikely that adaptive and non-adaptive weight decay have fairly similar behavior.

In this section, we provide a mathematical demonstration for the convergence of AWD for a very simple convex optimization problem. Consider the following optimization problem:

$$\min_x MSE(x, \beta) = \min_x \frac{\|x - \beta\|^2}{2} \tag{13}$$

This problem is convex and even has a closed-form solution. However, let us consider the SGD solution with an adaptive weight decay regularizer:

$$\min_x MSE(x, \beta) = \min_x \frac{\|x - \beta\|^2}{2} + c\frac{|x - \beta|}{|x|} \tag{14}$$

where, $c = \frac{\lambda_{awd}}{2}$ and $x \neq 0$. We consider all cases for possible choices of $\beta$ and $x$ and show that for all 7 cases, as long as we chose the hyper-parameter $0 < c < \frac{\beta^2}{2}$, which translates to $0 < \lambda_{awd} < \beta^2$, the problem is locally convex in those regimes. In order to do that, we show that the second derivative is always positive in all cases. Table 12 summarizes all 7 cases and the conditions in which the second derivative is guaranteed to be positive.

| $\beta$ | $x$ | $\beta$ vs. $x$ | Minimization | Simplified Minimization | 2nd Deriv. | Conv. Cond. | $c$ Cond. |
|---|---|---|---|---|---|---|---|
| $\beta > 0$ | $x > 0$ | $x > \beta$ | $0.5\|x-\beta\|^2 + c\frac{\|x-\beta\|}{\|x\|}$ | $0.5(x-\beta)^2 + c\frac{x-\beta}{x}$ | $1 - \frac{2bc}{x^3}$ | $1 - \frac{2bc}{x^3} > 0$ | $c < \frac{b^2}{2}$ |
| $\beta > 0$ | $x > 0$ | $x < \beta$ | $0.5\|x-\beta\|^2 + c\frac{\|x-\beta\|}{\|x\|}$ | $0.5(x-\beta)^2 - c\frac{x-\beta}{x}$ | $1 + \frac{2bc}{x^3}$ | $1 + \frac{2bc}{x^3} > 0$ | $c > 0$ |
| $\beta > 0$ | $x < 0$ | $x < \beta$ | $0.5\|x-\beta\|^2 + c\frac{\|x-\beta\|}{\|x\|}$ | $0.5(x-\beta)^2 + c\frac{x-\beta}{x}$ | $1 - \frac{2bc}{x^3}$ | $1 - \frac{2bc}{x^3} > 0$ | $c > 0$ |
| $\beta < 0$ | $x > 0$ | $x > \beta$ | $0.5\|x-\beta\|^2 + c\frac{\|x-\beta\|}{\|x\|}$ | $0.5(x-\beta)^2 + c\frac{x-\beta}{x}$ | $1 - \frac{2bc}{x^3}$ | $1 - \frac{2bc}{x^3} > 0$ | $c > 0$ |
| $\beta < 0$ | $x < 0$ | $x > \beta$ | $0.5\|x-\beta\|^2 + c\frac{\|x-\beta\|}{\|x\|}$ | $0.5(x-\beta)^2 - c\frac{x-\beta}{x}$ | $1 + \frac{2bc}{x^3}$ | $1 + \frac{2bc}{x^3} > 0$ | $c > 0$ |
| $\beta < 0$ | $x < 0$ | $x < \beta$ | $0.5\|x-\beta\|^2 + c\frac{\|x-\beta\|}{\|x\|}$ | $0.5(x-\beta)^2 + c\frac{x-\beta}{x}$ | $1 - \frac{2bc}{x^3}$ | $1 - \frac{2bc}{x^3} > 0$ | $c < \frac{b^2}{2}$ |
| $\beta = 0$ | Any | - | $0.5\|x-\beta\|^2 + c\frac{\|x-\beta\|}{\|x\|}$ | $(0.5x-\beta)^2$ | $1$ | Always convex | Any $c$ |

Table 12: Conditions guaranteeing convexity and convergence of AWD for a simple regression problem.

As it can be seen from Table 12, $0 < c < \frac{\beta^2}{2}$ satisfies the local convexity condition for all 7 cases. As a result, adaptive weight decay formulation for this problem is always locally convex and given the right hyper-parameter for $\lambda_{awd}$, SGD always converges to a solution.