# OpenReview forum: "Improving Robustness with Adaptive Weight Decay"
_NeurIPS.cc/2023/Conference — NeurIPS 2023 poster_

### Official Review · Reviewer_3gZW · 2023-07-03

**Soundness:** 2 fair
**Presentation:** 2 fair
**Contribution:** 2 fair
**Rating:** 6
**Confidence:** 3

**Summary:**

To improve the robustness, this paper proposes a method to determine the weight decay hyper-parameter during adversarial training adaptively. The key idea is to select the proper weight decay parameter $(\lambda_t )$ to keep the decay over the gradient (DoG) as a constant. The proposed method is evaluated on image classification and label noises tasks, comparing with the adversarial training with fixed weight decay hyper-parameter $(\lambda_{wd})$. According to the experimental results, adversarial training with dynamic $\lambda_t$ performs better.

**Strengths:**

1) The comparison results visualized by figures (Figures 2 & 3) are concise and clear.

**Weaknesses:**

1) The writing is not good, and the research gap and motivation are vague.
2) How this method will benefit pruning, as mentioned in the abstract, is not well supported.
3) The essential part of determining the constant value of DoG is not clearly stated.

**Questions:**

1) The way to organize the paper does not look good to me. The introduction part (lines 16 - 37) is too short, and the related work section (lines 149 - 155) appears after the proposed method. The relations between the proposed work and those existing works are blurred. I cannot see a clear clue about how the idea came up or what bottlenecks this proposed method aims to solve.
2) About the grid search strategy for $\lambda_{wd}$ and the selection for $DoG$. Taking Figure 3 as an example, I am curious about how to search for those two hyper-parameters. What is the search interval, and what is the search step? Will the search stage be very costly? Can there be a more efficient guide to deciding the constant value of DoG?
3) About the generality of this method. The proposed method can only be utilized with cross-entropy plus $l_2$-norm regularizer? If yes, I am concerned about the generality of this proposed method.
4) About the benefits that this method can bring to network pruning. This point is mentioned in the abstract (line 14) and Section 3 (line 219). However, this paper does not provide any mathematical analysis or experimental results. More important, the listed references (lines 218 - 219) are not state-of-the-art. Thus, I think the claim is a bit exaggerated.
5) About the editorial issues that make the reading difficult. The proposed method is called Adaptive Weight Decay / AWD for a while and DoG for a while. I suggest using a fixed abbreviation for the proposed method. Besides, many descriptions are inconsistent, though not big issues from the grammar aspect, they harm the reading experience significantly:
- when mentioning a section, lines 31 & 35, lines 146 & 212, and lines 190 & 195. There are at least three different ways to mention the section.
- when mentioning a figure, line 88 and line 99 are inconsistent.
- when defining a variable, the definitions of $w$ are different in line 45 and line 68, parameter or parameters?
- when mentioning the algorithm, lines 127 and line 133 are different.
- …

---

> ### Author Rebuttal · Authors · 2023-08-09
>
> We thank reviewer 3gZW for their insightful comments and great editorial suggestions. We will incorporate these suggestions in the final version of the paper. Please review our response to some of the questions you asked.
>
> > How this method will benefit pruning, as mentioned in the abstract, is not well supported.
>
> We thank you for this comment. Please note that we are not claiming state-of-the-art or even on-par performance for pruning. However, to incorporate your feedback, we have changed the language used in our manuscript regarding the pruning to reflect our claims more accurately.
>
> We claim there is more potential for pruning once the network is trained with adaptive weight decay than the non-adaptive method. The intuition for this is due to our observations in the pruning experiments. Unfortunately, we moved the pruning experiments to the appendix in favor of preserving space for more critical/relevant experiments in the main body of the paper. But here is a brief overview of the experiments: We train a WRN28-10 with both the adaptive and non-adaptive methods on the CIFAR100 dataset with various values for learning rate. The results show that the adaptive trained network outperforms the non-adaptive method, regardless of the learning rate used to train each network. Then we prune 70% of the variables of each of these networks with a non-iterative L1-norm based method and evaluate the accuracy of the networks after pruning. The adaptive trained networks still outperform the non-adaptive networks. The results are summarized in Figure 9 in the appendix.
>
>
> > More important, the listed references (lines 218 - 219) are not state-of-the-art. Thus, I think the claim is a bit exaggerated.
>
> We want to clarify that the two references mentioned in lines 218-219 are the earliest research that propose the weight decay method for the first time. We cite them as the original papers that proposed the method, and we do not use them as a baseline for comparison.
>
>
> > About the generality of this method. The proposed method can only be utilized with cross-entropy plus l2-norm regularizer? If yes, I am concerned about the generality of this proposed method.
>
> In the paper, we focus on this very common setting. However, we have noticed some colleagues get inspired by the robustness benefits of this adaptive formulation and its use case for balancing different terms in the objective and utilizing it in other settings such as localization.
>
>
> > About the grid search strategy for λwd and the selection for DoG. Taking Figure 3 as an example, I am curious about how to search for those two hyper-parameters. What is the search interval, and what is the search step? Will the search stage be very costly? Can there be a more efficient guide to deciding the constant value of DoG?
>
> We are glad you asked this question and would like to thank you for asking. In our earliest experiments, we performed a joint grid search on the learning rate and weight decay hyper-parameters to study the common properties of well performing sets of hyper-parameters. We monitored different statistics like the norm of weights, the norm of gradients, etc. We plotted the ratio between the norm of gradients coming from weight decay to the norm of gradients coming from the main loss. We observed that experiments that performed better had similar values for this metric, which we later called DoG. We conducted another experiment that would enforce the DoG value to be a constant during the entire training, which we later renamed Adaptive Weight Decay.
>
> To answer your question, in general, tuning the hyper-parameter of AWD will result in a better result; however, if, due to limited resources, it is infeasible to conduct a grid search or if the question is how to decide the initial range for the grid search, assuming that we know a good set of hyper-parameters for the non-adaptive method, we can estimate a well-performing hyper-parameter for the adaptive method. We suggest training the network with the non-adaptive (WD) method and monitor the average DoG during the training until the optimization converges. Based on our experience, the average DoG will be a very close to and a fair approximation for the best-performing DoG hyper-parameter after the grid search.
>
> To prove this point, we leveraged this method to find a good hyper-parameter for the non-adaptive method and reproduced results similar to that of Rebuffi et al. [1] with the adaptive method. We trained a WRN-28-10 with Swish activation, with a batch size of 1024 for 800 epochs, with the extra data similar to the experiments in Table 2 of Rebuffi et al. Since sweeping the hyper-parameters would require considerable resources, we used the hyper-parameters explained in Rebuffi et al. as our hyper-parameter for WD. To estimate a good hyper-parameter for AWD, we monitored the average(DoG) value during the training of the non-adaptive method. Table G1 summarizes the comparison between WD and AWD with extra data.
>
> |Name|Lambda|Natural Acc|AutoAttack|
> |:-:|:-:|:-:|:-:|
> |Rebuffi et al.|WD=0.0005|89.42|63.05|
> |Rebuffi + AWD|AWD=0.18|**90.53**|**63.55**|
>
> Table G1: Performance of AWD with additional data.
>
> For all experiments through out the paper and the rebuttal, we used the average(DoG) of WD experiment (i.e. 0.02) to get an estimate for the optimum DoG. Then we used a geometric progression with 16 steps between 2x smaller (i.e. 0.01) and 2x larger (i.e. 0.04) values of the estimated DoG for our grid search.
>
>
> > About the editorial issues that make the reading difficult. The proposed method is called Adaptive Weight Decay / AWD for a while and DoG for a while.
>
> Thanks for this suggestion. We strongly believe that your suggestion can improve the readability of our manuscript and we will update the next version of manuscript to address this issue.
>
> [1] Rebuffi, S.A, et al. “Fixing data augmentation to improve adversarial robustness”. arXiv:2103.01946

---

> ### Comment · Reviewer_3gZW · 2023-08-20
> **Response to the Authors**
>
> Thanks for the detailed explanations about my questions about: 1) the relations between the proposed method and pruning and, 2) the hyper-parameter selection strategy. Based on that, I would like to raise my score to 6.

---

### Official Review · Reviewer_pbDj · 2023-07-04

**Soundness:** 4 excellent
**Presentation:** 4 excellent
**Contribution:** 3 good
**Rating:** 5
**Confidence:** 3

**Summary:**

This paper proposes a simple but efficient way to improve model robustness: Adaptive Weight Decay, which automatically tunes the hyper-parameter for weight decay during each training iteration. Experimental results prove that this method significantly improves the robustness of the model on multiple datasets.

**Strengths:**

1. The paper is well-written and the proposed methods are clearly formulated.
2. Experimental results prove that this method significantly improves the robustness of the model on multiple datasets.

**Weaknesses:**

1. It would be better if more theoretical explanations about AWD could be provided.

2. Although the proposed method has a great improvement over the baseline method, it does not compare with other SOTA methods. In particular, on CIFAR10, the AWD only achieves 50.03% robustness under AA attack using the WideResNet-28-10. As far as I know, it is less robust than other methods (e.g., early stop).

3. The hyperparameter (DoG) of AWD has a great influence on the result, which makes the proposed method not so adaptive.

**Questions:**

See Weaknesses

**Limitations:**

See Weaknesses

---

> ### Author Rebuttal · Authors · 2023-08-09
>
> We thank you, reviewer pbDj, for your constructive criticism and insightful feedback. Please review our response to the concerns raised.
>
>
> > It would be better if more theoretical explanations about AWD could be provided.
>
> We absolutely agree with your point. A theoretical analysis would add more value to the paper. In this empirical paper, we tried our best to explain and verify the properties of our method by performing thorough experimental analysis and have shown significant benefits of using AWD for robustness despite its simplicity. We agree that theoretical analysis of this method is an interesting topic for future works.
>
> > Although the proposed method has a great improvement over the baseline method, it does not compare with other SOTA methods. In particular, on CIFAR10, the AWD only achieves 50.03% robustness under AA attack using the WideResNet-28-10. As far as I know, it is less robust than other methods (e.g., early stop).
>
> Thank you for your suggestion regarding comparison with other SOTA methods on CIFAR-10. Please note that Table 1 from the original paper was mainly used as a motivation table, and while we used validation-based early stopping for both conventional WD and AWD, we had not explored other methods, nor had we done experiments on architectures used for reporting CIFAR-10 numbers in many of the SOTA papers. As you requested, we have conducted experiments on WRN 32-10 (the common choice for running SOTA experiments) and have summarized our results in the Table P1. This Table has the same format as Table 2 in the main body but focuses on comparisons to more advanced methods on the CIFAR-10 dataset.
>
> |Dataset|Method|WRN|Aug|Epochs.|Nat|AutoAttack|
> |:-:|:-:|:-:|:-:|:-:|:-:|:-:|
> |CIFAR-10 (New)|AT (Madry et al., 2017)|32-10|P&C|100|87.80|48.46|
> |CIFAR-10 (New)|TRADES (Zhang et al., 2019)|32-10|P&C|100|86.36|53.40|
> |CIFAR-10 (New)|MART (Wang et al., 2020)|32-10|P&C|100|84.76|51.40|
> |CIFAR-10 (New)|FAT (Zhang et al., 2020a)|32-10|P&C|100|**89.70**|47.48|
> |CIFAR-10 (New)|AWP (Wu et al., 2020)|32-10|P&C|100|57.55|53.08|
> |CIFAR-10 (New)|GAIRAT (Zhang et al., 2020b)|32-10|P&C|100|86.30|40.30|
> |CIFAR-10 (New)|MAIL-AT (Liu et al., 2021)|32-10|P&C|100|84.83|47.10|
> |CIFAR-10 (New)|MAIL-TR (Liu et al., 2021)|32-10|P&C|100|84.00|53.90|
> |CIFAR-10 (New)|AWD (ours) with Dog=0.022 + ASAM|32-10|P&C|100|88.55|**54.04**|
>
> Table P1: CIFAR-10 robustness comparisons
>
> As it can be seen, despite its simplicity, our method outperforms many great algorithms on the CIFAR-10 dataset both in terms of natural accuracy and robustness. We will also update our manuscript's next version to include these results.
>
>
> > The hyperparameter (DoG) of AWD has a great influence on the result, which makes the proposed method not so adaptive.
>
> We do agree that correctly tuning this hyper-parameter, similar to many other hyper-parameters, can result in more favorable results. However, we note that the robustness/performance of AWD is less sensitive and easier to tune compared to the traditional weight-decay hyper-parameter, lambda (e.g., please see Fig. 11 in the appendix.) The comparatively lower sensitivity has enabled us to re-use the D0G hyper-parameter (0.022) found before for various experiments during the rebuttal period with different settings (See Table R1 and Table R2). This hyper-parameter value has been working fine and has resulted in comparable performances to the tuned hyper-parameter. Please see for example the Table P2 which we use the same setting as that of Table R2 and report the robust accuracy for both the optimal DoG parameter and the fixed value (0.022) for various choices of epochs.
>
> |Dataset|DoG|Epochs.|Nat|AutoAttack|
> |:-:|:-:|:-:|:-:|:-:|
> |CIFAR-100|Re-use (0.022)|50|62.85|29.25|
> |CIFAR-100|Optimize (0.024)|50|61.43|29.40|
> |CIFAR-100|Re-use (0.022)|100|64.49|29.70|
> |CIFAR-100|Optimize (0.024)|100|62.72|29.82|
> |CIFAR-100|Re-use (0.022)|150|64.17|29.94|
> |CIFAR-100|Optimize (0.022)|150|64.17|29.94|
> |CIFAR-100|Re-use (0.022)|200|64.37|29.55|
> |CIFAR-100|Optimize (0.022)|200|64.37|29.55|
> |CIFAR-100|Re-use (0.022)|250|63.24|29.68|
> |CIFAR-100|Optimize (0.022)|250|63.24|29.68|
> |CIFAR-100|Re-use (0.022)|300|63.35|29.28|
> |CIFAR-100|Optimize (0.024)|300|61.03|29.56|
>
> Table P2: AWD is not sensitive to its hyper-parameter.

---

> > ### Comment · Reviewer_pbDj · 2023-08-16
> > **Reply to the author**
> >
> > Thanks for the authors' response. After reading the rebuttals, I still have some concerns.
> >
> > 1 Please describe the experimental setup in Figure 3 in detail. it is confusing.
> >
> > 2 From Figure 3 of the submitted manuscript, the hyperparameter (DoG) of AWD has a great influence on the result, If we face a new dataset or network, how should we choose DoG?

---

> > > ### Author Response · Authors · 2023-08-16
> > >
> > > > 1 Please describe the experimental setup in Figure 3 in detail. it is confusing.
> > >
> > > Thank you, dear Reviewer pbDj, for helping us identify that the plotting function did not sort the x-axis in figure3. We believe it may have been the source of confusion? While the results remain intact, we will fix the plotting in the camera-ready/future versions. Since we could not update the figure during the discussion period, we have gathered the information of that figure and compiled it in table format (and included clean/nat accuracy) where the hyper-parameters are sorted (Table1-Table4).
> > >
> > >
> > >
> > > We would also like to use this opportunity to clarify the experiments in Figure 3. We train WRN28-10 networks using traditional weight-decay (with lambda hyperparameter) and another time using AWD (with DoG hyperparameter). In these experiments, we vary the hyper-parameters [by doing a grid search within a reasonable range] and do validation-based-early-stopping to pick the most robust checkpoint using the held-out validation set. We then report the robust accuracy on the test set by attacking the models using AutoAttack (AA). Regarding the range of the hyperparameters searched, we search for twice as many hyperparameters for WD compared to AWD and ensure that the optimal point is not at one of the two ends of the search range.
> > >
> > > |DoG|C10 - Nat|C10 - AA|C100 - Nat|C100 - AA|
> > > |:-:|:-:|:-:|:-:|:-:|
> > > |0.02|**88.05**|49.01|**62.85**|25.75|
> > > |0.02181|87.08|**50.03**|61.39|**27.15**|
> > > |0.02378|86.78|49.72|59.51|26.61|
> > > |0.02594|85.90|50.00|55.97|26.61|
> > > |0.02828|84.50|49.21|51.12|25.70|
> > > |0.03084|82.25|47.93|47.24|24.63|
> > > |0.03364|77.15|45.53|41.16|22.69|
> > > |0.03668|72.91|44.23|36.30|20.64|
> > > |0.04|67.56|41.35|31.03|18.19|
> > >
> > > Table 1: Table from Figure 3 -- AWD for CIFAR10 and CIFAR-100.
> > >
> > > |Lambda|C10 - Nat|C10 - AA|C100 - Nat|C100 - AA|
> > > |:-:|:-:|:-:|:-:|:-:|
> > > |0.00005|42.62|82.66|20.86|54.90|
> > > |0.000067|41.97|83.26|19.37|53.95|
> > > |0.000089|43.43|81.89|20.39|59.91|
> > > |0.000119|43.39|82.27|21.08|59.50|
> > > |0.000158|43.28|82.71|21.51|59.45|
> > > |0.000211|43.38|85.97|21.66|59.37|
> > > |0.000281|43.64|82.11|21.90|60.01|
> > > |0.000375|44.19|86.34|21.75|59.61|
> > > |0.0005|44.03|86.40|**22.53**|60.15|
> > > |0.00067|44.24|**86.59**|21.61|60.54|
> > > |0.00089|**45.19**|84.31|21.55|**61.33**|
> > > |0.00119|43.00|84.29|21.19|57.75|
> > > |0.00158|44.15|85.17|21.29|59.31|
> > > |0.00211|43.39|85.11|22.06|59.99|
> > > |0.00281|43.35|85.34|22.07|58.26|
> > > |0.00375|42.27|81.31|20.49|50.66|
> > > |0.005|37.23|72.96|17.76|41.96|
> > >
> > > Table 2: Table from Figure 3 -- WD for CIFAR-10 and CIFAR-100
> > >
> > > |Lambda|Tiny - Nat|Tiny - AA|
> > > |:-:|:-:|:-:|
> > > |5E-05|45.08|10.44|
> > > |6.7E-05|45.15|10.72|
> > > |8.9E-05|45.81|10.86|
> > > |0.000119|46.51|10.81|
> > > |0.000158|43.81|10.75|
> > > |0.000211|44.89|11.25|
> > > |0.000281|47.04|11.94|
> > > |0.000375|48.08|11.58|
> > > |0.0005|49.77|11.79|
> > > |0.00067|50.24|11.79|
> > > |0.00089|50.01|12.41|
> > > |0.00119|**52.90**|15.08|
> > > |0.00158|52.05|16.42|
> > > |0.00211|47.87|**16.73**|
> > > |0.00281|38.85|14.25|
> > > |0.00375|24.46|8.41|
> > > |0.005|6.67|2.70|
> > >
> > > Table 3: Table from Figure 3 -- WD for TinyImageNet
> > >
> > > |Dog|Tiny - Nat|Tiny - AA|
> > > |:-:|:-:|:-:|
> > > |0.01361|**54.35**|17.33|
> > > |0.0147|53.11|18.79|
> > > |0.01587|51.34|19.38|
> > > |0.01714|48.46|**19.74**|
> > > |0.01852|45.23|19.74|
> > > |0.02|41.90|19.33|
> > > |0.02181|36.34|17.68|
> > > |0.02378|30.93|15.74|
> > > |0.02594|25.97|13.89|
> > >
> > > Table 4: Table from Figure 3 -- AWD for TinyImageNet
> > >
> > > > 2 From Figure 3 of the submitted manuscript, the hyperparameter (DoG) of AWD has a great influence on the result, If we face a new dataset or network, how should we choose DoG?
> > >
> > > Thank you for raising this question. Given that a similar question was also raised by reviewer 3gZW, we will include a reference to this in the appendix for the future revision. While as seen in figure3, the hyper-parameter value of 0.021 has resulted in improvements over the optimal hyper-parameter for traditional weight decay across all 3 datasets, the results can be improved even further by doing a proper grid search.
> > >
> > > The preferred method for finding the best hyper-parameter for adaptive weight decay (i.e., DoG) is to treat it as any other hyper-parameter and perform a grid-search. But having a good estimate for the range used during grid search has value. Hence, we present a method to find a good estimate which works well in practice. We suggest that if a training pipeline exists that is fully tuned for non-adaptive/traditional weight decay, we can get a reasonable estimate for DoG by taking an average (or moving average) of the DoG values from all iterations during training with traditional weight decay. This estimate works well in practice for training with AWD and comes with almost no additional computation overhead.
> > >
> > > For most experiments through out the paper and the rebuttal, we used the average(DoG) of WD experiment (e.g. 0.02) to get an estimate for the optimum DoG. Then we used a geometric progression with 16 steps between 2x smaller (e.g. 0.01) and 2x larger (e.g. 0.04) values of the estimated DoG for our grid search.

---

> > > > ### Comment · Reviewer_pbDj · 2023-08-21
> > > > **Reply to the author**
> > > >
> > > > Thanks for the authors' response.  From the experimental results provided by the author, I am getting more and more confused. For example, in Table 2: Table from Figure 3 -- WD for CIFAR-10 and CIFAR-100, the author claimed the network is WRN28-10. the robustness of the WRN28-10 under AA is too lower.  Please refer to the results in [1]. I think the experimental settings of this result may be unreasonable, resulting in the low results of Table 2.
> > > >
> > > > Secondly, how to select DoG still needs to go through a grid search, which makes the proposed algorithm not adaptive.
> > > >
> > > >
> > > > Ref: [1]  BAG OF TRICKS FOR ADVERSARIAL TRAINING

---

> > > > > ### Author Response · Authors · 2023-08-21
> > > > >
> > > > > Thank you for sharing your feedback and allowing us to clarify the numbers in the motivational/baseline figure3 (more specifically, Table2 in our rebuttal.) Table2 uses common settings for training (w/o) any extra optimizations/modifications apart from tuning weight-decay and uses validation-based-early stopping per Rice et al. We continue using this common setting for our method in the SOTA comparison table (table P1) as well. Please note that in Table P1, where we have compared to SOTA methods, the robustness numbers for CIFAR-10 are higher than the best reported ones in [1] which use more similar architectures. For Table P1 we use a WRN32-10 which is fairly similar to WRN34-10 used in [1]. Please note that after all the tricks used in Table9 of [1], the most robust WRN34-10 model under the 8/255 adversarial attack has 53.88 AA robust accuracy and 84.24 clean accuracy, while when training a WRN32-10 using AWD — using conventional training settings and without any bag of tricks — our model has 54.04 robust accuracy and 88.55 clean accuracy.
> > > > >
> > > > > In regards to Table2 in the rebuttal and the training setting used in [1]: Training setting in [1] is an interesting setting which differs from the work of rice et al. and many others. The most notable difference is the learning rate schedule, where in [1] it is tailored to work well on CIFAR-10 such that robust overfitting is remedied to a great extent. To be more precise, in [1], they train for 110 epochs and use an initial learning rate of 0.1, which drops to .01 at the 100th epoch, and further drops to 0.001 at the 150th epoch. This is while, in all works that discuss robust overfitting, the learning rate either follows a cosine or it drops to 0.01 halfway through training, and it further drops to 0.001 during the last quarter of training. We would like to clarify that in all of our experiments we use a training setting that is a common setting used by most works in the literature that report results on various datasets. We use this common setting for Table2 in our rebuttal response as well to demonstrate the effectiveness of adaptive weight-decay over traditional weight-decay to prevent robust overfitting. While redoing Table2 for CIFAR-10, with the training setting used in [1] might still be interesting, given that only a few hours remain in the discussion period and, also that our AWD models in Table P1 (with much less bag of tricks) are both more robust and have higher clean accuracy than that reported in [1] (with many bag of tricks + uncommon training setting), we would like to believe that AWD still demonstrates its usability in settings which suffer from robust overfitting.

---

### Official Review · Reviewer_61Q2 · 2023-07-05

**Soundness:** 2 fair
**Presentation:** 2 fair
**Contribution:** 2 fair
**Rating:** 4
**Confidence:** 4

**Summary:**

This paper proposed adaptive weight decay which is balancing the gradient of the loss funciton such as the coss-entropy loss and the weight decay term.
Althogh the proposed method is a simple method, it empirically improves  adversarial robustness and a classification with label noise.

**Strengths:**

The strong point  of this paper is the simplicity of the proposed method.
The proposed method is just  adaptively rebalancing the gradient of the loss and weight decay term.
It empiciallly outperforms the existing studies of adversarial robustness.

**Weaknesses:**

The main concern with this method is that we do not know what the algorithm is ultimately optimizing.
We cannot know if the algorithm converges even in optimization problems such as convex optimization.
We also cannot know why this method is good for adversarial robustness and label noize.

**Questions:**

Q1. I would like to know if this algorithm is designed to minimize the original optimization problem.
It is difficult to set up in general, but usually when considering adaptive optimization algorithms like this, it is important for the safety of the algorithm that convergence is guaranteed, at least for optimization of convex functions.

Q2. What is inevitable about equation (7).
Of course, you are scaling with the gradient of the cross entropy loss, but I would like to know the direct effect of why that is effective for adversarial robustness.

Q3. Why do robust validation accuracy and loss suddenly improve around 200 epochs?
Since it is possible that the performance is good when stopping early at a good 200 epochs by chance, we would like to see a discussion on convergence.

Q4. What happens if we increase the number of epochs up to 300 epochs?

Q5. In Table 2, Epo varies for existing methods; if each method is adjusted in terms of early stopping, this may not be a problem, but the conditions should be as consistent as possible.

Q6. Does the proposed method work well in a more natural setting such as ImageNet-A?
The adversarial robustness setting is a bit artificial, which is a problem of this fields itself.

Q7. What is the performance in the normal problem setting instead of adversarial robustness?
In particular, do we see a performance decrease with different DOG settings?
If so, how do we determine the final DOG value?
 Even if the user makes the final decision, if the performance of the normal setting is significantly degraded as a result of considering adversarial robustness, we may not want to use it as a method.

**Limitations:**

No convergence guarantees for adaptive algorithms.
No theoretical basis for performance improvement against adversarial robustness.

---

> ### Author Rebuttal · Authors · 2023-08-09
>
> We thank reviewer 61Q2 for their insightful comments. Please review our response to questions asked.
>
> > The main concern with this method is that we do not know what the algorithm is ultimately optimizing. We cannot know if the algorithm converges even in optimization problems such as convex optimization. We also cannot know why this method is good for adversarial robustness and label noize.
>
> We have tried our best to explain our intuition in section 2.2.1. As suggested by the literature, regularizers prevent overfitting and overfitting hurts generalization of DNNs on unseen data. We experimentally showed that AWD is a stronger regularizer, compared to the non-adaptive method. We believe less overfitting is helping the resulting network to have a better performance on both adversarial robustness and label noise.
>
> > Q1. I would like to know if this algorithm is designed to minimize the original optimization problem. It is difficult to set up in general, but usually when considering adaptive optimization algorithms like this, it is important for the safety of the algorithm that convergence is guaranteed, at least for optimization of convex functions.
>
> As an empirical paper, we verified the effectiveness of this method through various experiments. While our final formulation might seem more complicated, our method can be re-written as:
> $$ min_{w}: Loss(Net_{w}(x), y) + C_{i} \|w\|_1 $$
> Which is very similar to weight decay, with L1 norm penalized instead of L2 norm. The main difference between AWD and non-adaptive method is that the regularization hyper-parameter in the L1 formula is changing for every iteration in AWD.
>
> Intuitively, we argue that if for a convex loss function $Loss$ and any constant $C$, the L1 formulation is a convex optimization problem, then at every step of our algorithm, we are solving a convex problem as well. In practice we see that the fluctuations in $C_{i}$ are rather small and in practice, all experiments that we have launched to this day have successfully converged.
>
> > Q2. What is inevitable about equation (7). Of course, you are scaling with the gradient of the cross entropy loss, but I would like to know the direct effect of why that is effective for adversarial robustness.
>
> The adaptive reformulation (scaling) can result in: 1) easier optimization of both objectives in the main loss in comparison to that of traditional weight-decay as evident by the results in the last column of Table 1; 2) possibility of reaching solutions with smaller weight-norms that could be seen as flatter minima which generalize better; 3) and most importantly, alleviating robust overfitting by allowing for stronger regularizations.
>
> > Q3 and Q4. Why do robust validation accuracy and loss suddenly improve around 200 epochs? Since it is possible that the performance is good when stopping early at a good 200 epochs by chance, we would like to see a discussion on convergence. What happens if we increase the number of epochs up to 300 epochs?
>
> That is a good observation. To empirically address the question about convergence, we have conducted a set of experiments where we keep everything constant and only vary the number of epochs for training. As it can be seen in Table R2 (in general response), and Table P2 (response to reviewer pbDj), AWD is not very sensitive to the number of training epochs. Our method always outperforms the non-adaptive method, both in terms of natural and adversarial accuracy.
>
> > Q5. In Table 2, Epo varies for existing methods; if each method is adjusted in terms of early stopping, this may not be a problem, but the conditions should be as consistent as possible.
>
> Thanks for bringing this to our attention. Initially, we had used the common setting from Madry et al., and had trained with 200 epochs. We have included 100 epochs in Table R3 (general response). AWD models trained with 100 epochs have similar performance to those trained with 200 epochs.
>
> > Q6. Does the proposed method work well in a more natural setting such as ImageNet-A? The adversarial robustness setting is a bit artificial, which is a problem of this fields itself.
>
> Given the limited time and available resources during the rebuttal period, we could not perform ImageNet-A experiments. However, ImageNet results are in the appendix. We are not sure if our method will be helpful for settings where under-fitting is happening. ImageNet is a dataset that is hard to fit, hence the adaptive and non-adaptive method perform similarly. We believe that both methods (traditional and adaptive weight decay) should have similar performance on ImageNet-A.
>
> > Q7. What is the performance in the normal problem setting instead of adversarial robustness? In particular, do we see a performance decrease with different DOG settings? If so, how do we determine the final DOG value? Even if the user makes the final decision, if the performance of the normal setting is significantly degraded as a result of considering adversarial robustness, we may not want to use it as a method.
>
> Finding a solution which improves both natural accuracy and adversarial robustness is one of the main goals of any robust algorithm. Throughout our adversarial training experiments, AWD results in solutions which improve both natural and robust accuracy under various hyper-parameters and datasets compared to traditional weight-decay.
>
> In regards to using AWD with different DoGs outside of the context of adversarial robustness, since in the normal problem settings overfitting is less of an issue, we mainly see on-par performance between AWD and traditional WD when the hyperparameters for each method is appropriately tuned. For example, please see Fig. 11 in appendix. However, we have noticed that even in the normal settings AWD has some robustness benefits over WD such as robustness to learning-rate as evident in Fig. 11 which could make hyperparameter searches easier.

---

> > ### Comment · Reviewer_61Q2 · 2023-08-18
> > **After Rebuttal**
> >
> > Thanks for your reply to my review.
> >  I have cleared up some questions, but I still have questions about the following two points.
> >
> > >Q1. I would like to know if this algorithm is designed to minimize the original optimization problem. It is difficult to set up in general, but usually when considering adaptive optimization algorithms like this, it is important for the safety of the algorithm that convergence is guaranteed, at least for optimization of convex functions.
> >
> > What I want to know is not an intuitive explanation, but what is actually being solved as an optimization problem and does the proposed algorithm converge in a simple convex minimization.
> >
> > > Q2. What is inevitable about equation (7). Of course, you are scaling with the gradient of the cross entropy loss, but I would like to know the direct effect of why that is effective for adversarial robustness.
> >
> > You explains this reason  in terms of flatter minima and generalize.
> > However, flatness is a formulation based on perturbations in parameter space, while adversarial robustness is a formulation based on perturbations in input space.
> > Thus, this is not a direct explanation.

---

> > > ### Author Response · Authors · 2023-08-20
> > >
> > > We are happy that many of your concerns have been addressed and are grateful for giving us an opportunity to address/clarify the 2 remaining ones. Please find our responses below.
> > >
> > >
> > > > Q1. What I want to know is not an intuitive explanation, but what is actually being solved as an optimization problem and does the proposed algorithm converge in a simple convex minimization.
> > >
> > >
> > > Following your ask, below, we show that for a simple choice of convex functions, our proposed loss function is locally convex (at least near the minimizer), and hence SGD will be able to converge to a stable/stationary local minima.
> > >
> > > Consider the following simple regression problem where the main objective is MSE: $0.5 \cdot \| x- \beta \|^2$. In this setting, our total loss formulation that includes the adaptive weight-decay regularizer term will be: $0.5 \cdot \| x- \beta \|^2 + c \cdot \frac{|x-\beta|}{|x|}$, where $c=0.5 \cdot DoG$ and $x \neq 0$. We consider all cases for possible choices of $\beta$ and $x$ and show that for all 7 cases, as long as we chose a the hyper-parameter $0 < c < \frac{\beta^2}{2}$ which translates to $ 0 < DoG < \beta^2 $ , the problem is locally convex in those regimes by showing that the second derivative is always positive.
> > >
> > >
> > > |$\beta$|$x$|$\beta$ vs. $x$ Comparison|Minimization|Simplified Minimization|2nd Derivative|Condition for Convexity|$c$ Satisfying Convexity|
> > > |:-:|:-:|:-:|:-:|:-:|:-:|:-:|:-:|
> > > |$\beta>0$|$x>0$|$x>\beta$|$0.5\cdot\|x-\beta\|^2+c\frac{ \|x-\beta\| }{ \|x\| }$|$0.5\cdot(x-\beta)^2+c\frac{x-\beta}{x}$|$1-\frac{2\beta c}{x^3}$|$1-\frac{2\beta c}{x^3}>0$|$c<\frac{\beta^2}{2}$|
> > > |$\beta>0$|$x>0$|$x<\beta$|$0.5\cdot\|x-\beta\|^2+c\frac{ \|x-\beta\| }{ \|x\| }$|$0.5\cdot(x-\beta)^2-c\frac{x-\beta}{x}$|$1+\frac{2\beta c}{x^3}$|$1+\frac{2\beta c}{x^3}>0$|$c>0$|
> > > |$\beta>0$|$x<0$|$x<\beta$|$0.5\cdot\|x-\beta\|^2+c\frac{ \|x-\beta\| }{ \|x\| }$|$0.5\cdot(x-\beta)^2+c\frac{x-\beta}{x}$|$1-\frac{2\beta c}{x^3}$|$1-\frac{2\beta c}{x^3}>0$|$c>0$|
> > > |$\beta<0$|$x>0$|$x>\beta$|$0.5\cdot\|x-\beta\|^2+c\frac{ \|x-\beta\| }{ \|x\| }$|$0.5\cdot(x-\beta)^2+c\frac{x-\beta}{x}$|$1-\frac{2\beta c}{x^3}$|$1-\frac{2\beta c}{x^3}>0$|$c>0$|
> > > |$\beta<0$|$x<0$|$x>\beta$|$0.5\cdot\|x-\beta\|^2+c\frac{ \|x-\beta\| }{ \|x\| }$|$0.5\cdot(x-\beta)^2-c\frac{x-\beta}{x}$|$1+\frac{2\beta c}{x^3}$|$1+\frac{2\beta c}{x^3}>0$|$c>0$|
> > > |$\beta<0$|$x<0$|$x<\beta$|$0.5\cdot\|x-\beta\|^2+c\frac{ \|x-\beta\| }{ \|x\| }$|$0.5\cdot(x-\beta)^2+c\frac{x-\beta}{x}$|$1-\frac{2\beta c}{x^3}$|$1-\frac{2\beta c}{x^3}>0$|$c<\frac{\beta^2}{2}$|
> > > |$\beta=0$|Any|-|$0.5\cdot\|x-\beta\|^2+c\frac{ \|x-\beta\| }{ \|x\| }$|$0.5\cdot(x-\beta)^2$|1|Always convex|Any $c$|
> > >
> > > Table R1: Conditions guaranteeing convexity and convergence of AWD for a simple regression.
> > >
> > >
> > >
> > >
> > >
> > > > Q2. flatness is a formulation based on perturbations in parameter space, while adversarial robustness is a formulation based on perturbations in input space. Thus, this is not a direct explanation.
> > >
> > >
> > > We would like to clarify this possible confusion that we did not mean that the adversarial training could yeild flatter minima, and we completely agree with the reviewer. Our response was in regards to the question about why in terms of adversarial robustness, AWD-trained models perform better compared to those trained with traditional weight decay. To elaborate further, we associate this to the model parameters/weights. As seen in Table 1 from the original paper (we have gathered the most relevant subset below in Table R2), AWD models have significantly smaller weight-norms compared to the best models trained with traditional weight decay. Models with smaller weight-norms are often simpler and could generalize better (as long as the model is still capable of fitting the data).
> > >
> > >
> > > |Dataset|Method|$\|W\|_2$| Nat | Adv|
> > > |:-:|:-:|:-:|:-:|:-:|
> > > |CIFAR-10|Weight-Decay|35.58|84.31|45.19|
> > > |CIFAR-10|AWD|**7.11**|**87.08**|**50.03**|
> > > |CIFAR-100|Weight-Decay|51.32|60.15|22.53|
> > > |CIFAR-100|AWD|**13.41**|**61.39**|**27.15**|
> > > |Tiny ImageNet|Weight-Decay|25.62|47.87|16.73|
> > > |Tiny ImageNet|AWD|**15.01**|**48.46**|**19.74**|
> > > |SVHN|Weight-Decay|102.11|92.04|44.16|
> > > |SVHN|AWD|**5.39**|**93.04**|**47.10**|
> > > |FashionMNIST|Weight-Decay|14.39|83.96|78.73|
> > > |FashionMNIST|AWD|**9.05**|**85.42**|**79.24**|
> > > |Flowers|Weight-Decay|19.94|**90.98**|32.35|
> > > |Flowers|AWD|**13.87**|90.39|**39.22**|
> > >
> > > Table R2: Relevant subset of columns from Table 1 which show that AWD trained models have considerably smaller weight-norms compared to models trained with traditional weight decay which results in simpler models which generalize better.

---

### Official Review · Reviewer_N3sV · 2023-07-06

**Soundness:** 3 good
**Presentation:** 4 excellent
**Contribution:** 3 good
**Rating:** 6
**Confidence:** 5

**Summary:**

The paper proposes adaptive weight decay (AWD) to adaptively tune the weight decay hyperparameter during training. The AWD keeps the ratio of weight decay update and cross-entropy loss update constant for the stability of training. The experiment shows that AWD improves adversarial robustness and reduces robust overfitting in adversarial training.

**Strengths:**

The paper is clearly written and easy to follow.

The experiment shows the benefit of AWD over several baselines on CIFAR10/100 and Tiny ImageNet.

**Weaknesses:**

As a reviewer who reviewed the submission for 3 times, my major concern before is the comparison of AWD with recent baselines like MART and MAIL. I am glad that Table 2 of this version includes such a comparison. The result looks promising to me so I would not raise any major weaknesses for this version.

One minor issue is the training time difference between DoG and baselines. Table 2 shows that DoG needs 200 epochs of training while all baselines use 100 epochs. It is not clear how these baselines will perform if we increase the training time, with learning rate decay delayed of course.

**Questions:**

Is the proposed method still better if baselines like MART and MAIL are trained for the same time as AWD?

**Limitations:**

Not discussed in the main paper.

---

> ### Author Rebuttal · Authors · 2023-08-09
>
> We thank you, reviewer N3sV, for your insightful comments. Please review our following feedback.
>
>
> > As a reviewer who reviewed the submission for 3 times, my major concern before is the comparison of AWD with recent baselines like MART and MAIL. I am glad that Table 2 of this version includes such a comparison. The result looks promising to me so I would not raise any major weaknesses for this version.
>
> Thank you for all your great suggestions throughout this paper's journey. Your constructive comments and thoughts have shaped this paper into what it is today, and we would like show our warmest appreciation and thank you for it.
>
>
> > One minor issue is the training time difference between DoG and baselines. Table 2 shows that DoG needs 200 epochs of training while all baselines use 100 epochs. It is not clear how these baselines will perform if we increase the training time, with learning rate decay delayed of course. Is the proposed method still better if baselines like MART and MAIL are trained for the same time as AWD?
>
> Thank you for bringing this to our attention. Initially, for training our models, we followed the conventional 200-epochs-training of robust CIFAR models (from [1]). But comparing methods in terms of their convergence speed by keeping the number of epochs constant is also very valuable. We have ran experiments of our method with keeping the DoG hyper param fixed to 0.022 and varying the number of epochs. The results are summarized Table N1.
>
> |Dataset|Method|WRN|Aug|Epochs.|Nat|AutoAttack|
> |:-:|:-:|:-:|:-:|:-:|:-:|:-:|
> |CIFAR-100|AWD (ours) with Dog=0.022 + ASAM|32-10|P&C|5|26.79|11.03|
> |CIFAR-100|AWD (ours) with Dog=0.022 + ASAM|32-10|P&C|10|38.10|15.92|
> |CIFAR-100|AWD (ours) with Dog=0.022 + ASAM|32-10|P&C|20|51.27|21.67|
> |CIFAR-100|AWD (ours) with Dog=0.022 + ASAM|32-10|P&C|30|58.10|25.03|
> |CIFAR-100|AWD (ours) with Dog=0.022 + ASAM|32-10|P&C|40|62.01|27.51|
> |CIFAR-100|AWD (ours) with Dog=0.022 + ASAM|32-10|P&C|50|62.85|29.25|
> |CIFAR-100|AWD (ours) with Dog=0.022 + ASAM|32-10|P&C|100|**64.49**|29.70|
> |CIFAR-100|AWD (ours) with Dog=0.022 + ASAM|32-10|P&C|150|64.17|**29.94**|
> |CIFAR-100|AWD (ours) with Dog=0.022 + ASAM|32-10|P&C|200|64.37|29.55|
> |CIFAR-100|AWD (ours) with Dog=0.022 + ASAM|32-10|P&C|250|63.24|29.68|
> |CIFAR-100|AWD (ours) with Dog=0.022 + ASAM|32-10|P&C|200|63.35|29.28|
>
> Table N1: CIFAR-100 WRN32-10 AWD varying epochs for studying convergence
>
> While at 100 epochs, our results (both nat and robustness) are comparable to our models trained with 200 epochs, we found that even if we further reduce the epochs to 50, we do not see a big degradation of robust accuracy, although the natural accuracy degrades slightly. When we go below 50 epochs, we see that the robustness accuracy degrades. Also in in the Table R2 in the general response, we have done a comparison between WD and AWD for various choices of number of epochs. In Table R2, for the WD case we have tuned the lambda hyper-parameter and picked the best one (the one that results in the highest robust accuracy), as opposed to the AWD, where we only report the results for the fixed hyper-parameter DoG=0.22.
>
> We have also updated the Table 2 from our original manuscript to include the results of our method with 100 epochs. For your reference, Table N2 summarizes the results.
>
> |Dataset|Method|WRN|Aug|Epochs.|Nat|AutoAttack|
> |:-:|:-:|:-:|:-:|:-:|:-:|:-:|
> |CIFAR-100|AT (Madry et al., 2017)|32-10|P&C|100|60.13|24.76|
> |CIFAR-100|TRADES (Zhang et al., 2019)|32-10|P&C|100|60.73|24.90|
> |CIFAR-100|MART (Wang et al., 2020)|32-10|P&C|100|54.08|25.30|
> |CIFAR-100|FAT (Zhang et al., 2020a)|32-10|P&C|100|**66.74**|20.88|
> |CIFAR-100|AWP (Wu et al., 2020)|32-10|P&C|100|55.16|25.16|
> |CIFAR-100|GAIRAT (Zhang et al., 2020b)|32-10|P&C|100|58.43|17.54|
> |CIFAR-100|MAIL-AT (Liu et al., 2021)|32-10|P&C|100|60.74|22.44|
> |CIFAR-100|MAIL-TR (Liu et al., 2021)|32-10|P&C|100|60.13|24.80|
> |CIFAR-100|AWD (ours) with Dog=0.022 + ASAM|32-10|P&C|200|64.37|29.55|
> |**CIFAR-100 (New)**|**AWD (ours) with Dog=0.022 + ASAM**|**32-10**|**P&C**|**100**|64.49|**29.70**|
>
> Table N2: CIFAR-100 SOTA Comparisons w/o additional data and same number of epochs
>
>
>
> [1] Madry, Aleksander, et al. "Towards deep learning models resistant to adversarial attacks." arXiv preprint arXiv:1706.06083 (2017).
>
> [2] Rebuffi, S.A, et al. “Fixing data augmentation to improve adversarial robustness”. arXiv preprint arXiv:2103.01946.

---

> > ### Comment · Reviewer_N3sV · 2023-08-16
> > **After Rebuttal**
> >
> > Thanks to the authors for consistently improving this paper along the (kind of long) journal. My concerns are addressed and I will keep my score.

---

### Official Review · Reviewer_c9pV · 2023-07-13

**Soundness:** 3 good
**Presentation:** 3 good
**Contribution:** 3 good
**Rating:** 6
**Confidence:** 4

**Summary:**

This paper studies the overfitting phenomena that is known to happen during adversarial training, with focus in image classification. The main idea hinges on the fact that weight regularization can be an effective technique to prevent such overfitting. The authors propose to augment the regular cross-entropy objective during training with a weight-decay regularization with an adaptive parameter to better utilize this toward preventing overfitting. Due to lack of theoretical analysis, the idea is tested numerically with empirical test on a number of image classification tasks and its effectiveness is shown empirically.

**Strengths:**

The main advantage of this work is its simplicity as well as it adequate development of the idea from initial observations to intuitively explaining why it makes sense to propose this approach.

**Weaknesses:**

The tests carried out in this paper seem somewhat limited. While the idea is based on state-of-the-art work in the literature [Rice et al., etc.], the results are not compared with these algorithms. Different choices of $\epsilon$ as well as attacks are also needed to show effectiveness of the proposition across a wide range of setups, especially given that the work is a heuristic approach without any theoretical evidence.

**Questions:**

1- The tests should be carried out for a couple of different values of $\epsilon$ as well as other standard attacks, to demonstrate performance across a range of settings. In addition to AutoAttack, performance against PGD attacks with large enough #steps (say K= 40 or 100) and small enough step-size (say $2 \epsilon /K)$ is common practice to report.

2- Results in table 1 need to also be compared with other state of the art algorithms for comparison, and not just constant versus adaptive weight decay.

3- As mentioned in the paper, using additional data has shown to be effective in improving robustness. How effective is adaptive decay if also combined with such additional data?

4- denoting the gradient of the cross-entropy loss part of the objective with respect to $w_t$ as $\nebla w_t$ is mathematically wrong (see eq. 5 , 6 ,7). This needs to be corrected for mathematical soundness. Also, it is reflected in line 6 of the Algorithm 1. Instead, one should define a function, say $F(.)$ to denote the cross entropy, and use that for mathematical soundness.

**Limitations:**

Yes

---

> ### Author Rebuttal · Authors · 2023-08-09
>
> We thank you, reviewer c9pV for your insightful comments. Please review our response to some of the questions you asked.
>
>
> > While the idea is based on state-of-the-art work in the literature [Rice et al., etc.], the results are not compared with these algorithms.
> Thank you for the suggestion regarding comparison with stronger baselines.
>
> We do agree about the significance of the work of (Rice et al.) In all of our experiments, we do validation-based early stopping; hence, our conventional WD experiments can be seen as the results from Rice et. al.  In addition to that, we compare our CIFAR-100 results to several other SOTA methods in Table 2 of the original submission. During the rebuttal period, we have also included comparisons to SOTA methods on CIFAR-10 which are summarized in Table C1 below:
>
> |Dataset|Method|WRN|Aug|Epochs.|Nat|AutoAttack|
> |:-:|:-:|:-:|:-:|:-:|:-:|:-:|
> |CIFAR-10 (New)|AT (Madry et al., 2017)|32-10|P&C|100|87.80|48.46|
> |CIFAR-10 (New)|TRADES (Zhang et al., 2019)|32-10|P&C|100|86.36|53.40|
> |CIFAR-10 (New)|MART (Wang et al., 2020)|32-10|P&C|100|84.76|51.40|
> |CIFAR-10 (New)|FAT (Zhang et al., 2020a)|32-10|P&C|100|**89.70**|47.48|
> |CIFAR-10 (New)|AWP (Wu et al., 2020)|32-10|P&C|100|57.55|53.08|
> |CIFAR-10 (New)|GAIRAT (Zhang et al., 2020b)|32-10|P&C|100|86.30|40.30|
> |CIFAR-10 (New)|MAIL-AT (Liu et al., 2021)|32-10|P&C|100|84.83|47.10|
> |CIFAR-10 (New)|MAIL-TR (Liu et al., 2021)|32-10|P&C|100|84.00|53.90|
> |CIFAR-10 (New)|AWD (ours) with Dog=0.022 + ASAM|32-10|P&C|100|88.55|**54.04**|
>
> Table C1: CIFAR-10 robustness comparisons
>
>
> > Different choices of ϵ as well as attacks are also needed to show effectiveness of the proposition across a wide range of setups, especially given that the work is a heuristic approach without any theoretical evidence.
>
> We thank you for bringing this to our attention. As you requested, we have run experiments with several values for $\epsilon$. The experimental setup for these experiments is similar to that of Table 1 from the original paper, except for the difference in $\epsilon$ values used for training and evaluation. We run experiments on WRN28-10 for both CIFAR-10 and CIFAR-100. We use a fixed DoG hyper-param of 0.022, as opposed to non-adaptive WD, where we do a grid search and report the best WD per parameter setup according to robust test accuracy measured by conducting AA and PGD with various steps. Table R1 (in general response) shows that AWD outperforms conventional WD training for all epsilons in both natural accuracy and robustness.
>
> > 1- The tests should be carried out for a couple of different values of ϵ as well as other standard attacks, to demonstrate performance across a range of settings. In addition to AutoAttack, performance against PGD attacks with large enough #steps (say K= 40 or 100) and small enough step-size (say 2ϵ/K) is common practice to report.
>
> Thanks for this suggestion. As requested, we have included our robustness results with various PGD steps for adversarially trained WRN28-10 networks. We use the step size you suggested to evaluate the experiments summarized in Table R1 and Table R2 of our general response.
>
> We agree that our method should be tested against several attacks. We used AutoAttack as our primary evaluation metric since it incorporates four of the most well-known and strongest adversarial attacks, including APGD-CE (step size free PGD attack), APGD-DLR (step size free PGD attack with DLR loss), FAB (which minimizes the norm of the adversarial perturbation) (Croce & Hein, 2019), and SQUARE (which is a query efficient black box attack) (Andriushchenko et al., 2019). Our evaluations also show that AutoAttack results in the strongest attacks.
>
>
> > 2- Results in table 1 need to also be compared with other state of the art algorithms for comparison, and not just constant versus adaptive weight decay.
>
> Please note that the non-adaptive rows in Table 1 incorporate the early stopping method suggested by (Rice et al.), so they can be interpreted as a comparison to (Rice et al.). Per your request, we included our results to other methods on CIFAR10 in our response above (Table C1).
>
> > 3- As mentioned in the paper, using additional data has shown to be effective in improving robustness. How effective is adaptive decay if also combined with such additional data?
>
> As mentioned in our paper, our method is most effective in settings that suffer from [robust] over-fitting. If we have more training data to fit, less overfitting happens. As a result, our method is less likely to be as effective in such settings, and the gap between traditional WD and AWD tightens, and their robustness is more comparable.
>
> To further show our point, we train a WideResNet 28-10 with Swish activation, with a batch size of 1024 for 800 epochs, with the extra data similar to the experiments in Table 2 of (Rebuffi et al.). Since sweeping the hyper-parameters would require considerable resources, we used the hyper-parameters explained in (Rebuffi et al.) as our hyper-parameter for the WD experiment. To estimate a good hyper-parameter for the AWD experiment, we monitored the average (DoG) value during the training of the non-adaptive method. Table C2 compares the performance of WD and AWD with extra data. As seen in the table, the adaptive method achieves results comparable to the non-adaptive method, and the gap between the performance of the two methods is closing.
>
> |Name|Lambda/DoG|Natural Acc|AutoAttack|
> |:-:|:-:|:-:|:-:|
> |(Rebuffi et al.)|WD=0.0005|89.42|63.05|
> |(Rebuffi et al.) + AWD|AWD=0.18|**90.53**|**63.55**|
>
> Table C2: Performance of AWD with more data.
>
> > 4- correctness and fixes to formulations
>
> Thanks! We will update the next version of our manuscript to reflect your suggestions.

---

> > ### Comment · Reviewer_c9pV · 2023-08-21
> > **Response**
> >
> > I would like to thank the authors for their response. The above has addressed most of my concerns, and I have slightly increased my rating score.

---

### Author Rebuttal · Authors · 2023-08-08

Dear reviewers. Thanks for your insightful and constructive comments. Below, you may find tables mentioned in the detailed responses for each review. Due to character limitations, please find more detailed explanation of the following tables in the per-reviewer rebuttals.

|Eps|Data|Alg|Lambda/DoG|Nat|20|40|60|80|100|AA-SQ|AA-CE|AA-FAB|AA-T|Min|
|:-:|:-:|:-:|:-:|:-:|:-:|:-:|:-:|:-:|:-:|:-:|:-:|:-:|:-:|:-:|
|2|C10|WD|0.00089|94.2|83.2|83.1|83.2|83.1|83.1|86.9|82.7|82.7|82.5|82.5|
|2|C10|AWD|0.02181|**94.3**|**83.6**|**83.6**|**83.6**|**83.6**|**83.6**|**87**|**83.2**|**83.1**|**83**|**83**|
|2|C100|WD|0.00067|74.8|55.8|55.8|55.7|55.7|55.7|59.2|54.8|52.9|52.7|52.7|
|2|C100|AWD|0.02181|**75.2**|**56.7**|**56.7**|**56.7**|**56.6**|**56.7**|**59.6**|**56**|**53.7**|**53.4**|**53.4**|
|4|C10|WD|0.00158|91.7|70.7|70.7|70.5|70.6|70.6|75.3|69.2|69.4|69|69|
|4|C10|AWD|0.02181|**92**|**73.1**|**73.1**|**73.1**|**73**|**73**|**77.8**|**72**|**71.7**|**71.3**|**71.3**|
|4|C100|WD|0.00089|69.4|42|42|42|41.9|41.9|44.8|40.4|38.9|38.6|38.6|
|4|C100|AWD|0.02181|**71.5**|**46.8**|**46.7**|**46.7**|**46.7**|**46.7**|**48.3**|**45.2**|**41.2**|**40.8**|**40.8**|
|6|C10|WD|0.00119|88.7|59|58.9|58.9|58.9|58.9|63.9|56|56.5|55.8|55.8|
|6|C10|AWD|0.02181|**90**|**62.4**|**62.3**|**62.4**|**62.3**|**62.3**|**67.3**|**60.3**|**60**|**59.5**|**59.5**|
|6|C100|WD|0.00067|64.7|32.8|32.6|32.7|32.6|32.6|35|30.9|29.5|29.2|29.2|
|6|C100|AWD|0.02181|**66.3**|**39.6**|**39.5**|**39.5**|**39.5**|**39.5**|**39.8**|**37.7**|**33.4**|**33.1**|**33.1**|
|8|C10|WD|0.00158|84|49.5|49.2|49.4|49.4|49.4|54.3|46.5|45.1|44.7|44.7|
|8|C10|AWD|0.02181|**87.3**|**53.9**|**53.8**|**53.7**|**53.8**|**53.8**|**58.1**|**51.4**|**50.1**|**49.6**|**49.6**|
|8|C100|WD|0.00158|56.5|27.7|27.7|27.7|27.5|27.5|28.9|25.9|22.6|22.4|22.4|
|8|C100|AWD|0.02181|**61.6**|**33.1**|**33**|**33.1**|**33.1**|**33**|**32.5**|**31**|**26.7**|**26.4**|**26.4**|
|16|C10|WD|0.00119|70.4|32|31.7|31.6|31.5|31.7|30.5|27.1|22.6|21.6|21.6|
|16|C10|AWD|0.02181|**71.9**|**34**|**33.8**|**33.7**|**33.7**|**33.7**|**33.2**|**29.6**|**26.1**|**25.3**|**25.3**|
|16|C100|WD|0.00281|38.3|16.5|16.5|16.5|16.5|16.5|14.2|14.8|11.3|11|11|
|16|C100|AWD|0.02181|**41.5**|**19.8**|**19.7**|**19.6**|**19.5**|**19.5**|**17**|**17.7**|**13.9**|**13.4**|**13.4**|

Table R1: Performance of AWD vs. WD on adversarially trained WRN28-10 networks with various values for $\epsilon$.

|Arch.|Epoch|Data|Alg|Lambda/DoG|Nat|20|40|60|80|100|AA-SQ|AA-CE|AA-FAB|AA-T|AA|
|:-:|:-:|:-:|:-:|:-:|:-:|:-:|:-:|:-:|:-:|:-:|:-:|:-:|:-:|:-:|:-:|
|WRN28-10|50|C10|WD|0.00158|86.5|52.6|52.4|52.4|52.4|52.5|56.7|49.5|48.5|48.0|48.0|
|WRN28-10|50|C10|AWD|0.02181|**87.1**|**54.3**|**54.0**|**54.0**|**54.0**|**54.1**|**58.7**|**51.3**|**50.0**|**49.5**|**49.5**|
|WRN28-10|50|C100|WD|0.00211|59.5|29.8|29.7|29.7|29.7|29.6|31.1|28.0|25.3|25.1|25.1|
|WRN28-10|50|C100|AWD|0.02181|**61.9**|**32.4**|**32.4**|**32.4**|**32.4**|**32.3**|**32.8**|**30.5**|**26.7**|**26.4**|**26.4**|
|WRN28-10|100|C10|WD|0.00211|85.8|50.8|50.6|50.5|50.6|50.6|55.4|47.7|47.1|46.6|46.6|
|WRN28-10|100|C10|AWD|0.02181|**87.7**|**55.1**|**55.0**|**55.0**|**54.9**|**54.9**|**59.6**|**52.5**|**51.5**|**51.2**|**51.2**|
|WRN28-10|100|C100|WD|0.00281|58.2|28.2|28.1|28.0|28.1|28.1|29.4|26.3|23.7|23.4|23.4|
|WRN28-10|100|C100|AWD|0.02181|**62.5**|**33.3**|**33.3**|**33.3**|**33.2**|**33.2**|**33.0**|**31.2**|**27.1**|**26.7**|**26.7**|
|WRN28-10|200|C10|WD|0.00158|84.1|50.3|50.1|50.1|50.0|50.0|54.8|47.4|46.2|45.7|45.7|
|WRN28-10|200|C10|AWD|0.02181|**87.3**|**54.2**|**54.0**|**54.0**|**54.0**|**54.0**|**58.5**|**51.4**|**50.5**|**50.0**|**50.0**|
|WRN28-10|200|C100|WD|0.0005|60.5|25.2|25.2|25.1|25.0|25.1|27.0|23.1|22.4|22.2|22.2|
|WRN28-10|200|C100|AWD|0.02181|**62.0**|**33.1**|**32.9**|**33.0**|**32.9**|**32.9**|**32.1**|**30.9**|**26.8**|**26.4**|**26.4**|
|WRN28-10|300|C10|WD|0.00089|86.2|48.9|48.8|48.6|48.6|48.6|52.8|44.9|45.2|44.5|44.5|
|WRN28-10|300|C10|AWD|0.02181|**87.3**|**52.8**|**52.7**|**52.8**|**52.8**|**52.7**|**57.4**|**50.3**|**49.4**|**48.8**|**48.8**|
|WRN28-10|300|C100|WD|0.00028|59.5|25.6|25.5|25.5|25.5|25.5|27.0|23.6|22.6|22.3|22.3|
|WRN28-10|300|C100|AWD|0.02181|**62.0**|**33.1**|**32.9**|**33.0**|**32.9**|**32.9**|**32.1**|**30.9**|**26.8**|**26.4**|**26.4**|

Table R2: Performance of AWD and WD on adversarially trained WRN28-10 trained for various epochs.

|Dataset|Method|WRN|Aug|Epochs.|Nat|AutoAttack|
|:-:|:-:|:-:|:-:|:-:|:-:|:-:|
|CIFAR-100|AT (Madry et al., 2017)|32-10|P&C|100|60.13|24.76|
|CIFAR-100|TRADES (Zhang et al., 2019)|32-10|P&C|100|60.73|24.90|
|CIFAR-100|MART (Wang et al., 2020)|32-10|P&C|100|54.08|25.30|
|CIFAR-100|FAT (Zhang et al., 2020a)|32-10|P&C|100|**66.74**|20.88|
|CIFAR-100|AWP (Wu et al., 2020)|32-10|P&C|100|55.16|25.16|
|CIFAR-100|GAIRAT (Zhang et al., 2020b)|32-10|P&C|100|58.43|17.54|
|CIFAR-100|MAIL-AT (Liu et al., 2021)|32-10|P&C|100|60.74|22.44|
|CIFAR-100|MAIL-TR (Liu et al., 2021)|32-10|P&C|100|60.13|24.80|
|CIFAR-100|AWD (ours) with Dog=0.022 + ASAM|32-10|P&C|200|64.37|29.55|
|CIFAR-100 (New)|AWD (ours) with Dog=0.022 + ASAM|32-10|P&C|100|64.49|**29.70**|
|CIFAR-10 (New)|AT (Madry et al., 2017)|32-10|P&C|100|87.80|48.46|
|CIFAR-10 (New)|TRADES (Zhang et al., 2019)|32-10|P&C|100|86.36|53.40|
|CIFAR-10 (New)|MART (Wang et al., 2020)|32-10|P&C|100|84.76|51.40|
|CIFAR-10 (New)|FAT (Zhang et al., 2020a)|32-10|P&C|100|**89.70**|47.48|
|CIFAR-10 (New)|AWP (Wu et al., 2020)|32-10|P&C|100|57.55|53.08|
|CIFAR-10 (New)|GAIRAT (Zhang et al., 2020b)|32-10|P&C|100|86.30|40.30|
|CIFAR-10 (New)|MAIL-AT (Liu et al., 2021)|32-10|P&C|100|84.83|47.10|
|CIFAR-10 (New)|MAIL-TR (Liu et al., 2021)|32-10|P&C|100|84.00|53.90|
|CIFAR-10 (New)|AWD (ours) with Dog=0.022 + ASAM|32-10|P&C|100|88.55|**54.04**|

Table R3: Comparison between AWD and other robustness methods.

---

### Decision · Program_Chairs · 2023-09-21

**Decision:**

Accept (poster)

**Comment:**

The paper investigates the overfitting that arises during adversarial training in image classification. The authors introduce adaptive weight decay (AWD) as a technique to prevent overfitting, maintaining a constant ratio between the weight decay update and cross-entropy loss update. Empirical experiments on multiple image classification tasks show the effectiveness of AWD over static decay.

### Strengths:
- Simplicity of the proposed method.
- Clear presentation and well-structured paper in most instances.
- Experimental results demonstrate the proposed method's superiority over several baselines on multiple datasets.

### Weaknesses:
- Lack of theoretical analysis to back up the approach.
- Some reviewers expressed concerns about the algorithm's optimization goals and whether it would converge, especially for convex optimization problems.
- Inadequate comparisons with certain state-of-the-art methods. Fortunately, this seems to be mitigated by new results in the rebuttal.
- Training times for the proposed approach and baselines seem inconsistent, leading to disparities in the experimental comparison. Fortunately, this seems to be mitigated by new results in the rebuttal.
- The method's hyperparameters, especially the DoG, appear to significantly impact results.
- Editorial inconsistencies and unclear presentation in certain sections, leading to confusion.

Considering the evident benefits of the proposed method against certain baselines and the simplicity of the approach, balanced against a lack of theoretical support and comparison against SOTA, Accept is recommended. However, authors should address the raised concerns, especially about the algorithm's convergence and improving clarity, in the manuscript. Please also incorporate the new results.